# Long-term Variances of Heavy Precipitation across Central Europe using a Large Ensemble of Regional Climate Model Simulations

Florian Ehmele[1], Lisa–Ann Kautz[1], Hendrik Feldmann[1], and Joaquim G. Pinto[1]

[1]Institute of Meteorology and Climate Research, Department Troposphere Research (IMK–TRO), Karlsruhe Institute of Technology (KIT), Hermann–von–Helmholtz–Platz 1, 76344 Eggenstein–Leopoldshafen, Germany.

**Correspondence:** Florian Ehmele (florian.ehmele@kit.edu)

**Abstract.** Widespread flooding events are among the major natural hazards in central Europe. Such events are usually related to intensive, long-lasting precipitation over larger areas. Despite some prominent floods during the last three decades (e.g. 1997, 1999, 2002, and 2013), extreme floods are rare and associated with estimated long return periods of more than 100 years. To assess the associated risks of such extreme events, reliable statistics of precipitation and discharge are required. Comprehensive observations, however, are mainly available for the last 50–60 years or less. This shortcoming can be reduced using stochastic data sets. One possibility towards this aim is to consider climate model data or extended reanalyses. This study presents and discusses a validation of different century-long data sets, decadal hindcasts, and also predictions for the upcoming decade combined to a new large ensemble. Global reanalyses for the 20th century with a horizontal resolution of more than 100 km have been dynamically downscaled with a regional climate model (COSMO–CLM) towards a higher resolution of 25 km. The new data sets are first filtered using a dry–day adjustment. Evaluation focuses on intensive widespread precipitation events and related temporal variabilities and trends. The presented ensemble data is within the range of observations for both statistical distributions and time series. The temporal evolution during the past 60 years is captured. The results reveal some long-term variability with phases of increased and decreased precipitation rates. The overall trend varies between the investigation areas but is mostly significant. The predictions for the upcoming decade show ongoing tendencies with increased areal precipitation. The presented RCM ensemble not only allows for more robust statistics in general, it is also suitable for a better estimation of extreme values.

## 1 Introduction

Ongoing climate change affects not only the global scale but also impacts the regional climate. Regarding air temperature, there is a more or less clear trend in the recent past, which reveals a clear anthropogenic signal. However, various climate simulations show distinct differences for precipitation trends, especially for heavy precipitation (e.g. Moberg et al., 2006; Zolina et al., 2008; Toreti et al., 2010). A review of observed variability and trends in extreme climate events states that it is difficult to find significant relations between the greenhouse gas-enhanced climate change and increases or decreases in extreme precipitation events (Field et al., 2012). This is attributed to their rare occurrence, the general high spatial variability of precipitation, and due to a lack of long-term high-quality observations.

Magnitude and sign of heavy precipitation trends strongly depend on various factors such as the regarded area or the considered time period (e.g. Easterling et al., 2000). Global tendencies towards more intense precipitation throughout the 20th century were revealed, for example, by Donat et al. (2016). Varying regimes between summer and winter season also account into precipitation trends. For example, Moberg and Jones (2005) found an increase in winter precipitation across central and western Europe between 1901 and 1999, while Pal et al. (2004) found a decrease in summer precipitation for the period 1951–

2000. Dittus et al. (2016) found an increasing trend between 1951 and 2005 in extreme total precipitation amounts for Europe in GCM simulations (CMIP5). Similar trends were found in global reanalyses (e.g. ERA–20C, Poli et al., 2016), but not in observations. In contrast, Primo et al. (2019) found positive trends for two ground-based observational stations in Germany using extreme precipitation indices.

    Model resolution is another crucial factor. The use of high resolution regional climate models (RCM) instead of global data sets

revealed a more detailed and orographically related spatial structure of the precipitation fields and trends (e.g. Feldmann et al., 2013). An increase of both areal mean precipitation and extremes in central Europe in order of 5–10 % was found in RCM simulations by Feldmann et al. (2013), which will continue with almost same magnitude for the next decade. Differences in precipitation trends also stem from varying definitions of extreme events such as certain thresholds, percentile-based indices, or return periods (e.g. Maraun et al., 2010). While most of these studies show trends in daily precipitation, just a few deal with

sub-daily trends. Barbero et al. (2017), for instance, compared trends in sub-daily and daily extremes. Although significant increasing trends were found for both time ranges, trends in daily extremes are better detected than in sub-daily extremes.

    Spatially extended intensive rainfall events are frequently related to widespread flooding along the main river networks of central Europe causing major damage in the order of several billion euro (EUR) per event (e.g. Uhlemann et al., 2010; Kienzler et al., 2015; Schröter et al., 2015; MunichRe, 2017). A prominent example of such an extreme and devastating event is the

flood in 2012 along the rivers Elbe and Danube (Ulbrich et al., 2003a, b). Such outstanding events are by definition extremely rare, which makes the risk estimation difficult or almost impossible due to the limited time period with available area-wide observations (e.g. Pauling and Paeth, 2007; Hirabayashi et al., 2013). However, trend analyses of such extreme events and the related risks during the past and for the future are of great importance for insurance purposes or flood protection (e.g. Merz et al., 2014; Schröter et al., 2015; Ehmele and Kunz, 2019).

A possible way of dealing with the unsatisfactory data availability are century-long simulations using climate models (e.g. Stucki et al., 2016) or stochastic approaches (e.g. Peleg et al., 2017; Singer et al., 2018; Ehmele and Kunz, 2019). The currently used GCMs were found to be in good agreement with the available but limited observations (Fischer and Knutti, 2016). Brönnimann et al. (2013) or Brönnimann (2017) analyzed historical extreme events using century-long reanalysis data sets and concluded that the quality of the reanalyses strongly depends on the number and type of the assimilated observations. The

investigated historical events were reproduced, but the magnitudes were underestimated. A possible reason is the decreasing number and quality of observations in the early century and therefore, a lack of assimilation data. The suitability of reanalysis data to investigate extreme precipitation for England and Wales was investigated by Rhodes et al. (2015). While time series of daily precipitation totals are well represented in both data sets, timing errors of heavy precipitation events were identified as one of the major problems. Stucki et al. (2012) investigated historical flooding events in Switzerland and indicate that the reanalyses

underestimate precipitation in Switzerland which may result from the insufficient representation of the alpine topography. The timing and the exact location of heavy precipitation were also found to be inaccurate.

As shown by van der Wiel et al. (2019) or Martel et al. (2020), large ensembles can have an added values for flood risk estimation and for the calculation of return periods of heavy precipitation. van der Wiel et al. (2019) found a clear benefit in using an ensemble approach for the estimation of changes in hydrological extremes including compound events compared to traditional approaches. Martel et al. (2020) found similar results, namely a reduction in the projected return period of 100-year annual maximum precipitation with the different ensembles, albeit having different model structures and resolutions. Furthermore, it was emphasized that a higher resolution is advantageous to predict climate change signals over complex terrain. Other studies also highlighted the improvements of using high resolution RCMs for the investigation of climate extremes (e.g. Feser et al., 2011; Feldmann et al., 2008, 2013; Schewe et al., 2019), especially over complex terrain (e.g. Torma et al., 2015).

The studies mentioned above document partly contrasting results and demonstrate the challenges arising when dealing with extreme precipitation and related phenomena. In this study, a set of different realizations with one RCM is used and combined to the new ensemble LAERTES-EU (**LA**rge **E**nsemble of **R**egional clima**T**e mod**E**l **S**imulations for **EU**rope), which can be used for more profound statistical analyses. Basis is the global reanalysis data set 20CR (Compo et al., 2011), which was dynamically downscaled for Europe. LAERTES-EU consists of a handful of 20th century reanalysis data sets and a large ensemble of decadal hindcast simulations mainly for the second half of the century. Although all simulations were performed with the same RCM version and set-up, LAERTES-EU is a combination of different external forcings, boundary conditions, and/or assimilation. Predictions for the upcoming decade will round up our analysis. The investigative focus lies on daily values of intensive areal precipitation which can be associated with major flood events in central Europe. As demonstrated for example by Schröter et al. (2015), severe flood events along the major river networks in central Europe are related to long-lasting and widespread precipitation events of mainly stratiform origin with embedded convective precipitation. Typically, intensities do not reach the most extreme rates of the distribution but are characterized by high spatial mean values.

LAERTES-EU is validated in terms of coincidence with observations regarding temporal variability, statistical distributions, and possible long-term trends. The following research questions will be addressed.

(1) How well is extreme areal precipitation represented in the RCM ensemble LAERTES-EU?

(2) What are potential benefits of LAERTES-EU compared to other available data sets?

(3) Which temporal evolution and variability of extreme areal precipitation over central Europe manifest during the past?

(4) Which tendency is expected for the upcoming decade?

A better interpretation of RCM data and a more profound understanding of extreme areal precipitation may have several applications such as risk assessments. Although being relevant, we do not handle the potential mechanisms behind temporal variances and trends as well as spatial and seasonal differences as this goes beyond the scope of this study.

This paper is structured as follows: The data sets which were used in this study are introduced in Sect. 2. Section 3 sums up the methods used for the analysis and the validation. In Sect. 4 LAERTES-EU is validated with observations for a reference

period. The investigation of temporal variabilities and trends is given in Sect. 5. Finally, Sect. 6 gives a summary and lists our conclusions.

## 2    Data sets

Two different types of data sets are applied in this study: gridded precipitation data based on observations and partly century-long climate model simulations (LAERTES-EU). The observational data sets are primarily available for the second half of the 20th century and serve as reference data for the validation of the ensemble. Furthermore, we compare LAERTES-EU with the forcing global model and also with the global reanalysis data set 20CR (Compo et al., 2011), which were used as initial data
for some of the simulations.

### 2.1    Observations

The European observational data set E–OBS version v17 including daily precipitation (Haylock et al., 2008; van den Besselaar et al., 2011) is a gridded data set with a horizontal resolution of $0.22°$ ($\approx 25\,\text{km}$) covering the years 1950 to 2017. This version shows some improvements towards older versions, since updated algorithms and new stations have been included in some areas
(e.g. for Poland). The E–OBS algorithm interpolates observations from weather stations to a regular grid using geostatistical methods (e.g. Journel and Huijbregts, 1978; Goovaerts, 2000). Note that E–OBS is a land-only data set and ocean grid points are set to a missing value. Haylock et al. (2008) stated that rainfall totals in E–OBS are reduced by up to almost one third compared to the raw station data at the corresponding grid cells. Regarding extremes, the deviation of E–OBS is even more pronounced (Hofstra et al., 2009). Nevertheless, both studies stated that the spatial mean precipitation in E–OBS is very close
to other observations.

Although E–OBS has some limitations, we use it as main reference for this study as there is no other comparable high-resolution daily precipitation data set available that covers entire Europe for a long time period. Other products like satellite data with a very limited time frame are not helpful and also have limitations. There are single ground-based observations with very long time series but as the focus of this study is on intensive areal precipitation this data is of limited usefulness for
validation.

Additionally to E–OBS, we compare the RCM simulations with the high-resolved HYRAS data set provided by the German Weather Service (DWD; Rauthe et al., 2013). HYRAS is a gridded precipitation data set with a horizontal resolution of up to 1 km for the time period 1951–2006 and covers Germany and the surrounding river catchments. The HYRAS algorithm also uses ground based measurements and interpolates the point observations to the regular grid. For this study, the HYRAS data
was first aggregated to the E–OBS/RCM 25 km grid. HYRAS hereafter means this aggregated 25 km data set.

### 2.2    Regional climate model simulations

LAERTES-EU combines a large number of regional dynamical downscaling simulations for Europe performed with a single RCM. The used RCM is the non-hydrostatic model of the Consortium for Small-scale Modelling (COSMO) in climate mode

model version 5 (CCLM5; Rockel et al., 2008), which has a spatial resolution of 0.22° (≈ 25 km). The model covers the EURO–CORDEX[1] domain (Jacob et al., 2014). Overall, the simulations use the same domain, model version and set-up, which was adapted from EURO–CORDEX (Kotlarski et al., 2014). According to Feldmann et al. (2008), a dry–day correction is important as climate models tend to overestimate the number of wet days with low intensities below 0.1 mm, known as the drizzle effect (Berg et al., 2012). In order to reduce this typical bias, a dry–day adjustment was first applied to LAERTES-EU. The E–OBS data were used for this correction, as they have the same spatial extension and resolution as the CCLM simulations. All simulations are performed within the BMBF (Federal Ministry of Education and Research of Germany) project MiKlip II[2] (Marotzke et al., 2016) to create and test a decadal prediction system including a regional downscaling component for Europe.

For all downscaling simulations the boundary conditions were derived from the Max–Planck Institute of Meteorology coupled Earth System Model (MPI–ESM). This global model consists of the atmospheric component ECHAM6 (Stevens et al., 2013), the ocean component MPI–OM (Jungclaus et al., 2013), and the land-surface model JSBACH (Hagemann et al., 2013).

LAERTES-EU is divided into four different data blocks (Table 1) depending on the setup of the forcing MPI–ESM ensemble simulations. The differences between the four data blocks stems from the setup, external forcing and initialization of the MPI–ESM simulations. The data blocks 1 and 2 of the RCM ensemble (cf. Table 1) obtained the boundary values from the MPI–ESM–LR simulations using a T63 resolution and 47 vertical layers. Data block 3 and 4 used the MPI–ESM–HR version (Müller et al., 2018) as their driving model. In this version, the horizontal resolution is T127 and 95 vertical layers are applied. Three types of forcing ensembles can be distinguished:

  (I)  MPI–ESM assimilates reanalysis data for long-term simulations (data block 1).

 (II)  Long-term historical-type simulations, according to the CMIP5 specifications (data block 3; Taylor et al., 2012).

(III)  Initialized decadal (10–year) hind- and forecast simulations (data blocks 2 and 4).

In data block 1, the first type (I) is applied. Here the 20th Century Reanalysis data (20CR; Compo et al., 2011) are assimilated into the MPI–ESM–LR (Müller et al., 2014). 20CR has a spatial resolution of approximately 2° (T62) and was generated using the Global Forecast System (GFS; Kanamitsu et al., 1991; Moorthi et al., 2001) of the National Centers for Environmental Prediction (NCEP)[3]. It used a 56 member Ensemble Kalman Filter approach to assimilate surface pressure, monthly sea surface temperature and sea-ice observations. Three of the 20CR members are assimilated into MPI–ESM to provide long-term (110 years each) climate reconstruction simulations over the period 1900–2009 (Müller et al., 2014). Afterwards, a downscaling with CCLM uses these global simulations as boundary conditions (e.g. Primo et al., 2019).

Data block 3 consists of the second type (II), were five so called historical simulations of MPI–ESM–HR with CMIP5 observed natural and anthropogenic external climate forcing (Taylor et al., 2012) are used as boundary conditions for CCLM. The ensemble was generated by starting the MPI–ESM from arbitrary dates in a pre-industrial control simulation (Müller et al.,

---

[1]http://www.euro-cordex.net

[2]https://www.fona-miklip.de/

[3]http://www.ncep.noaa.gov/

**Table 1.** Overview of the RCM ensemble LAERTES-EU with the name of the simulation within the MiKlip project, the classification into data blocks, the underlaying set-up (experiment), the covered time period, and the number of simulation years; XX stand for the ensemble number and YYYY for the initialization year.

| name | block | experiment | period | years | comment |
|------|-------|-----------|--------|-------|---------|
| as20ncep**XX** | 1 | 20CR via MPI–ESM–LR | 1900–2009 | 330 | 3 members of 110 years each |
| dec**XX**o**YYYY** | 2 | MPI–ESM–LR DROUGHTCLIP | 1911–2019 | 3000 | 3 members with 100 decades each |
| historical_r**X**i1p1-HR | 3 | MPI-ESM–HR HISTORICAL | 1900–2005 | 410 | run 1–3 each with 106 years, run 4–5 each with 46 years (1960–2005) |
| preop | 4 | MPI–ESM–HR CMIP5 | 1961–2026 | 2850 | 5 members with 57 decades each |
| dcppA-hindcast | 4 | MPI–ESM–HR CMIP6 | 1961–2028 | 5900 | 10 members with 59 decades each |

2014). Three of the five CCLM members cover the period 1900–2005 (106 years each). The two additional simulations cover the period 1960–2005 (46 years each).

Data block 2 and 4 consist of initialized decadal simulations (type III). The starting conditions are derived from an observed state (Müller et al., 2012; Marotzke et al., 2016). For each starting year, an ensemble of decadal simulations is generated and then, the initialization point is shifted by one year (e.g. 1961–1970, 1962–1971, and so on). Due to the overlap, a specific calendar year may be covered by several decadal hindcasts with different starting years. These decadal hind- and forecasts thus represent the current state of the major modes of climate variability compared to the so-called un-initialized historical simulations (data block 3). The downscaling procedure, the skill, and the added value are described in Mieruch et al. (2014), Feldmann et al. (2019), and Reyers et al. (2019).

In data block 2, the starting conditions of the three decadal hindcast members with MPI–ESM–LR are derived from the assimilation experiments in data block 1. The starting years of the CCLM downscaling range from 1910 to 2009. This means the last simulated year is 2019.

Data block 4 consists of two parts. Both of them use the MPI–ESM–HR version. The so-called preop-ensemble has five members. The external climate forcing is derived from CMIP5. The starting years range from 1960 to 2016 (last simulated year 2026). The so-called dcppA-hindcast ensemble has ten members and uses the external forcing for CMIP6 (Eyring et al., 2016). The global simulations are a contribution to the Decadal Climate Prediction Project of CMIP6 (DCPP; Boer et al., 2016). The starting years are 1960 to 2018 (last simulated year 2028).

In total, LAERTES-EU consists of 1183 simulation runs (sample size) with approximately 12.500 simulated years. The number of ensemble members for a specific year varies from six at the beginning of the century to a maximum of 188 members between 1970 and 2000 (see Fig. S1 in the supplemental material). The simulation in all four data blocks are affected by the observed external climate forcing, but they differ with respect to the representation of the observed climate variability,

whereas data block 1 uses assimilated 20CR reanalysis data, data block 2 and 4 contain initialized hindcasts, which to some degree follow the observed low frequency variability, and data block 3 only uses the external forcing information. Nonetheless, the four groups of downscaling simulations can be grouped into a large ensemble, since the regional simulations were all performed with the same setup of the RCM. Despite the same initial conditions and model setup, the temporal evolution of the day-to-day weather is (statistically) independent between the members after a few weeks. This is an advantage, since the data

set is homogeneous over time but also covers uncertainties in the observations including unknown and not yet observed events. The validity of this combination approach is tested within Sect. 4.

## 3 Methods

The capability of LAERTES-EU to simulate realistic precipitation amounts and distribution is an important requirement. Moreover, temporal variability and possible trends should also be well represented for trustworthy data sets. The methods were ap-

plied to different investigation areas and time periods. Equations and additional information can be found in Appendix A–C. As the focus of this study is intensive areal precipitation, we concentrate on high percentiles of spatially aggregated daily rainfall totals, namely 99 %, and 99.9 %. The percentiles are based on wet days only. First, a spatial aggregation of daily precipitation values was applied. Afterwards, the percentile of these areal precipitation were calculated for each year separately. In all data sets, ocean grid cells were set to a missing value and therefore neglected.

**3.1 Validation methods**

LAERTES-EU is analyzed and validated using various methods. The intensity spectrum gives the statistical probability of each precipitation amount by taking into account all grid points and all time steps within the investigation area and without any aggregation. Therefore, the range of occurred values is divided into evenly spaced histogram classes, which then are normalized with the total sample size. The resulting intensity–probability–curve (IPC) is a good indicator if the model is capable to simulate

realistic precipitation intensity distributions.

As an extension to the IPCs, the linear error in probability space $L$ (cf. Eq. A1–A3 in Appendix A) is analyzed (e.g. Ward and Folland, 1991; Potts et al., 1996). Therefore, empirical cumulative density functions (ECDF) are calculated for each simulation run and for the observations. The data basis is the same as for the IPCs. The value $\Delta C_r$ (Eq. A1) is defined as the difference between the ECDF of a model run $r$ and that of the observation (difference of probabilities) up to a specific precipitation

intensity. It is therefore a measure for the over- or underestimation of the model. Using $\Delta C_r$, the linear error in probability space ($L_r$; Eq. A2) is the mean of the absolute values $|\Delta C_r|$ over the entire precipitation range as defined by Déqué (2012) or Wahl et al. (2017). The better both density function coincide, the lower the value of $L_r$. According Eq. A2, $L_r$ is always positive. The ensemble mean is given by $\overline{L}$ (Eq. A3).

The internal variability of LAERTES-EU on different time intervals is compared to that of the observations. Given that the

focus of this study is on intensive widespread precipitation, this analysis is performed using spatial mean precipitation amounts averaged over the investigation areas. First, the time series of daily spatial means are aggregated over different intervals, namely

monthly, seasonal, and yearly precipitation sums as well as 5, 10, or 30-year running means. In a second step, the standard deviation of a gamma distribution $\sigma_\Gamma$ is calculated for each of these interval series (see Appendix A; Eq. A4), for every single member of LAERTES-EU, and for the observations. Finally, the ensemble mean of the four data blocks and of the complete ensemble is built. This method enables the analysis of how well the internal variability on different time scales is captured by LAERTES-EU.

The quantile–quantile (Q–Q) plot compares the simulated distribution with the observed one using different percentiles of daily spatial mean precipitation. The Q–Q distributions are used to calculate the coefficient of determination $R^2$ with $R$ being the Pearson correlation coefficient (Eq. A5).

The added value of the ensemble size is analyzed by using the signal–to–noise ratio $S2N$ (Eq. A6). Therefore, we determine a Gumbel distribution (cf. Appendix A) for different sample sizes and the corresponding 90 % confidence interval. The $S2N$ is then the ratio of the return value of the Gumbel distribution divided by the 90 % confidence interval (Früh et al., 2010).

### 3.2 Decadal variability and trend analysis

For the analysis of the temporal evolution of heavy precipitation, we use time series of different percentiles of spatial mean precipitation and quantities introduced and recommended by the Expert Team on Climate Change Detection and Indices (ETC-CDI; Karl et al., 1999; Peterson, 2005). Currently, 27 indices for temperature and precipitation are defined by the ETCCDI. These indices can be used from local to global scales. Additionally, they combine extremes with a mean climatological state (Zwiers et al., 2013). In this study, we use the two indices R95pTOT and R99pTOT (Eq. B1–B2 in Appendix B), which indicate the amount of precipitation above the 95 % or 99 % percentile, respectively.

In terms of trend analysis, a Mann–Kendall test (Mann, 1945; Kendall, 1955) is performed with related significance investigations (Appendix C). Regarding possible oscillations, the complete time series is split into sub-series with a minimum length of 10 years and up to 130 years (trend matrix). The Mann–Kendall test is applied to each of these sub-series.

### 3.3 Investigation areas and time periods

The focus of this study is central Europe, implying the countries Germany, Switzerland, the Netherlands, Belgium, Luxembourg, and parts of France, Poland, Austria, the Czech Republic, and Italy. Following Christensen and Christensen (2007), these countries are mostly coincident with two of the areas defined in the PRUDENCE project (prediction of regional scenarios and uncertainties for defining European climate change risks and effects), namely the PRUDENCE regions (PR) Mid–Europe (ME) and the Alps (AL; Fig. 1). Albeit these boxes contain both land and ocean, the latter was set to a missing value and neglected. During validation, ME and AL were reduced to the HYRAS grid cells lying within the corresponding box, hereafter referred to as ME$^*$ and AL$^*$.

The data sets are investigated on different time periods (TP): TP1 covers the past from 1900 to 2017, which is divided into a sub-period TP1b only containing the period 1951 to 2006, with both observations (E–OBS and HYRAS) being available. The time period TP2 is used for the predictions from 2018 to 2028. Note that the simulations were performed within the MiKlip project back in 2018 (using observations until 2017), which is the reason why the prediction period starts in 2018.

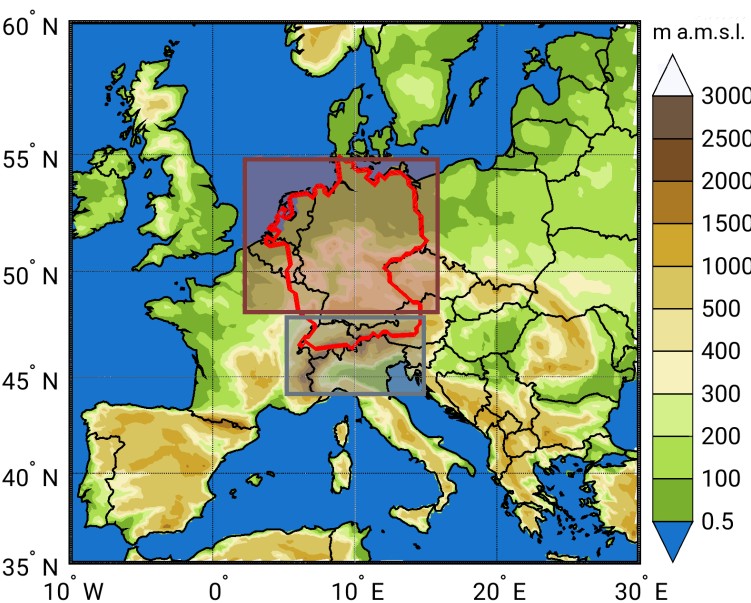

**Figure 1.** Topographic map of Europe at model resolution 0.22° (in meters above mean sea level; m a.m.s.l.) with the PRUDENCE regions Mid-Europe (ME; dark red box) and the Alps (AL; gray box), state borders (black contours), and the HYRAS area (light red contour). Ocean grid cells are set to a missing value.

For climatological aspects, we use the time period 1961–1990, hereafter referred to as climTP. A couple of studies (e.g. Cahill et al., 2015; Folland et al., 2018) showed that the climate change signal for global mean temperature significantly increased since the early 1980s. Therefore, using the time period 1981–2010 as reference would possibly include a strong changing signal to the analysis. Using 1961–1990 reduces the influence of these effects, as this period shows more stable conditions to a certain degree. This also permits more room for the interpretation of the future predictions.

## 4   Validation of the RCM ensemble

In the following, the above described methods are applied in order to validate LAERTES-EU concerning its representativeness with observations. With this aim, data for the investigation period TP1b is used and the boxes ME and AL (cf. Fig. 1) are limited to the HYRAS area (ME* and AL*).

### 4.1   Statistical distributions and frequencies

The IPCs give the range of simulated (observed) precipitation intensities at any grid point within the investigation area and its corresponding probability (Fig. 2). For both investigation areas, the IPCs reveal a distinct added value of the RCM compared to the global model. Due to the coarse resolution, intensities greater than approximately $100\,\mathrm{mm\,d^{-1}}$ are not found in the

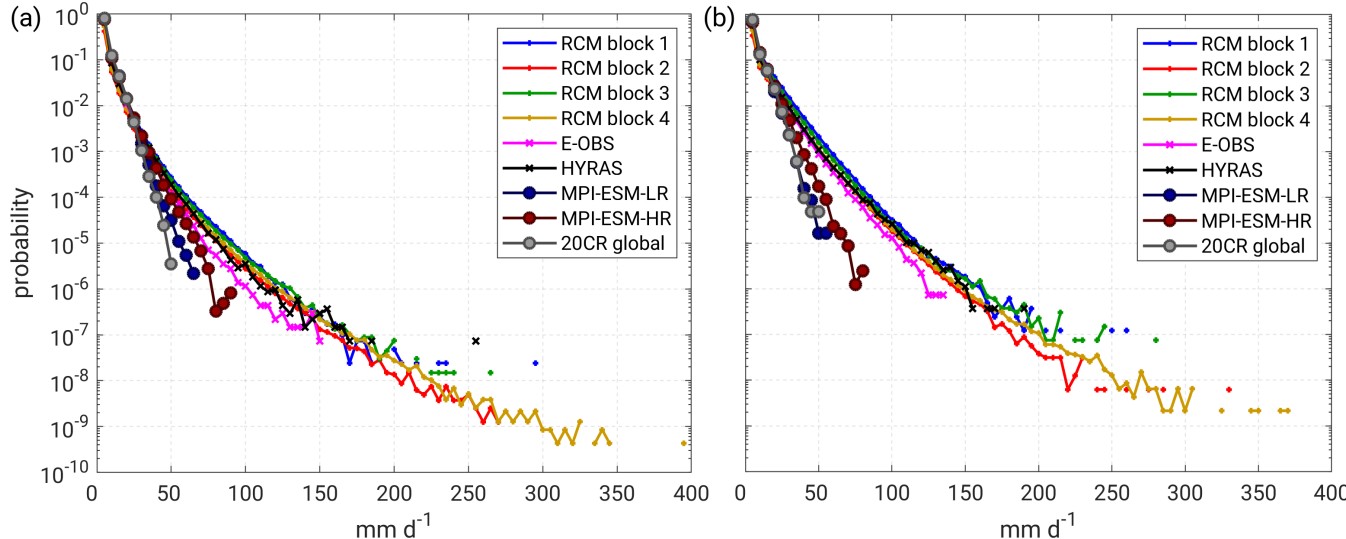

**Figure 2.** Intensity–probability–curves (IPCs) of daily rainfall totals of the RCM simulations (dry–day adjusted), observations (E–OBS and HYRAS), GCM simulations (forcing MPI–ESM data at two resolutions LR and HR), and global reanalysis data (20CR) for (a) Mid–Europe (ME*) and (b) the Alps (AL*), both limited to the HYRAS area during the investigation period TP1b (1951–2006). For the IPCs, every grid cell value at every time step was taken into account without any aggregation.

GCMs, which underestimate by a large degree the probability of the high intensities. The same applies for the global reanalysis 20CR. On the other hand, the RCM tends to overestimate the probability for precipitation intensities above a threshold of
approximately $50\,\mathrm{mm\,d^{-1}}$, but cover the entire range of values as the observations. The wider range of intensities at the upper tail of the distribution may include possibly not yet observed events.

For ME*, the IPCs of the RCM are close to HYRAS, but there is a systematic difference between HYRAS and E–OBS (Fig. 2a). As already mentioned by Haylock et al. (2008), E–OBS has a certain negative bias up to –30 % when using grid point based quantities. The given deviation of HYRAS and E–OBS is in between this range. Similar results can be found for
AL* (Fig. 2b). The differences between the RCM simulations and the observations at a given probability are slightly less than for ME*. For both areas the range of simulated values is much higher with up to $400\,\mathrm{mm\,d^{-1}}$. Naturally, higher intensities are more likely in the mountainous AL* region.

In contrast to the grid point based IPCs, Fig. 3 shows the mean standard deviation of a gamma distribution (cf. Sect. 3.1 and Appendix A) for the time series of spatial mean precipitation amounts aggregated over different time intervals. For both areas,
there is an expectable continuous decrease of internal variability towards longer periods for all data sets/data blocks. For ME*, LAERTES-EU is in good agreement with both observations at least up to a yearly perspective. For longer time periods, data block 1 shows a slightly different behavior compared to the other data blocks and observations. Nevertheless, data blocks 2–4 and the ensemble mean continue to match with the observations up to the 10-year running mean. Note that it is not possible to estimate the 30-year running mean for the decadal simulations of data block 2 and 4 given the data availability. For data

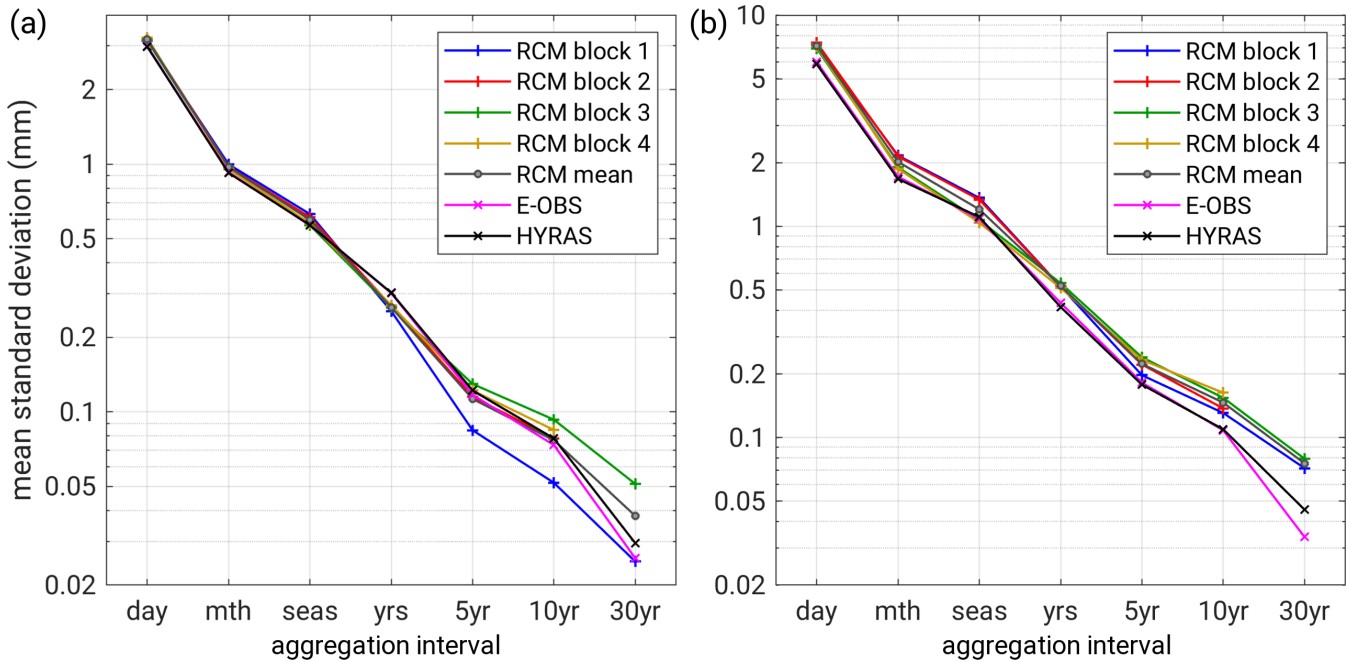

**Figure 3.** Mean standard deviation $\sigma_\Gamma$ in mm (mean over data blocks) of spatially averaged precipitation aggregated over different time intervals: daily (day), monthly (mth), seasonal (seas), yearly (yrs), and 5/10/30-year running mean (5yr/10yr/30yr) for (a) ME* and (b) AL* (TP1b; 1951–2006). The four data blocks of LAERTES-EU are considered separately; RCM mean stands for the complete ensemble mean (gray). The results for E–OBS and HYRAS are given in black and magenta. Note that it is not possible to estimate the 30yr values for the decadals of data blocks 2 and 4.

block3, only an external climate forcing was used meaning these so-called historicals are free runs in terms of daily weather evolution. Therefore, it is not expected that the multi-decadal variability is in phase to the observed circulation after a certain time, which can be a reason for slightly higher differences of data block 3 compared to the observations at the longest time scale. Furthermore, note that the results of Fig. 3 do not indicate a perfect match of LAERTES-EU in terms of absolute values, but rather that the internal variability (spread) of spatial mean precipitation totals is well captured. For the mountainous AL*

region, the internal variability is higher and all data blocks have a higher standard deviation at all time intervals. This means that the spread of simulated precipitation amounts is increased compared to that of the observation. A possible reason for this difference can emerge from sparse measurements in that region considered for both E–OBS and HYRAS, especially for long-term observations. The more or less constant difference between LAERTES-EU and the observations can be an indicator of a possibly systematic bias in this region.

The Q–Q plots of daily spatial mean precipitation fields for both investigation areas are shown in Fig. S2 in the supplemental material. Generally speaking, the distribution of the RCM is similar to those of the observations, at least to E–OBS, with little deviations from the optimum (diagonal line) for most of the spectrum and differences at around 10 % for the upper part of the distribution. In comparison to HYRAS, the maximum deviation is higher with around 20 %. For AL*, the differences between

**Table 2.** Coefficients of determination $R^2$ (top number) for the quantile–quantile contemplation of Fig. S2 and linear error in probability space $\overline{L}$ (bottom number) between the RCM and both observations (E–OBS and HYRAS) for Mid–Europe (ME$^*$) and the Alps (AL$^*$), always using HYRAS grid cells only. Both quantities are based on daily spatial mean precipitation amounts.

| RCM | E–OBS | | HYRAS | |
|---|---|---|---|---|
| | ME$^*$ | AL$^*$ | ME$^*$ | AL$^*$ |
| Data block 1 | 0.9914 | 0.9924 | 0.9876 | 0.9835 |
| | 0.0016 | 0.0058 | 0.0027 | 0.0080 |
| Data block 2 | 0.9914 | 0.9925 | 0.9878 | 0.9848 |
| | 0.0009 | 0.0037 | 0.0021 | 0.0058 |
| Data block 3 | 0.9963 | 0.9976 | 0.9936 | 0.9930 |
| | 0.0017 | 0.0062 | 0.0029 | 0.0083 |
| Data block 4 | 0.9966 | 0.9981 | 0.9943 | 0.9938 |
| | 0.0011 | 0.0038 | 0.0023 | 0.0059 |

the RCM and HYRAS are larger than for ME$^*$ (Fig. S2). Even though HYRAS was aggregated to the E–OBS/RCM grid, the more pronounced differences especially for the extremes might be a result of the higher resolution of the HYRAS data, which, in particular, is of greater relevance in the mountainous region of AL$^*$.

The findings of Fig. S2 are confirmed by the determination coefficients $R^2$ (Table 2). For both E–OBS and HYRAS, the coefficient is very high with $R^2 > 0.98$. There is a slightly higher $R^2$ for E–OBS than for HYRAS, which is an artificial effect of the data resolution. The region AL$^*$ shows a minimal higher skill compared to ME$^*$ in E–OBS and slightly lower values in HYRAS. Table 2 also reveals higher correlations of the CCLM simulations driven by the high-resolution MPI–ESM–HR data compared to those driven by the lower resolved MPI–ESM–LR data. Even though this seems to be systematic, the differences are marginal.

Table 2 also contains the mean linear error in probability space $\overline{L}$ for the different data blocks. Again, the differences between the data blocks are marginal with all cases being close to $\overline{L} = 0$ which stands for a good agreement of LAERTES-EU with observations. In contrast to $R^2$, $\overline{L}$ has lower values for the simulations driven by MPI–ESM–LR. For all data blocks, $\overline{L}$ is considerable higher for the mountainous AL$^*$ region. Note that both quantities being close to its optimum value does not indicate a perfect model. It rather means that the overall statistics regarding the entire range of intensities to a high degree coincide with the observations.

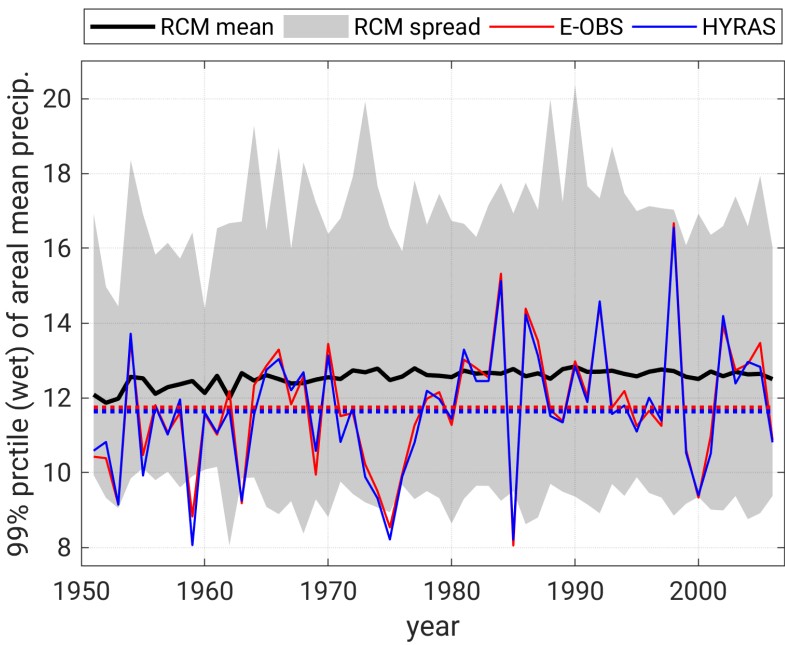

**Figure 4.** Time series of the yearly 99 % percentile (wet days and HYRAS area only) of daily spatial mean precipitation values for Mid–Europe (ME*) during TP1b (1951–2006) of the LAERTES-EU ensemble mean (black), the ensemble spread (minimum to maximum; gray), E–OBS (red), and HYRAS (blue). The dotted lines symbolize the mean values of the observations throughout TP1b.

## 4.2   Time series

Beside overall statistics, other properties of LAERTES-EU like the temporal variability should cover the range of observations as well. Therefore, we analyze the time series of yearly values of different percentiles of the spatial mean precipitation for the investigation areas. In Fig. 4, the time series of the 99 % percentile for ME* is shown. Both observational data sets have a high year–to–year variability with similar shape. The ensemble mean value of LAERTES-EU is higher, with a relative deviation of 1–10 % (TP1b average is 7 %). The spread of both observational data sets is covered by the ensemble spread (minimum to

maximum values) of LAERTES-EU except for few extreme peaks (e.g. 1985). In AL*, the E–OBS mean is about 5 % higher than HYRAS but both time series have again a similar shape (Fig. S3). The ensemble mean again is higher with relative deviations of 12–23 % (16 % on average) to E–OBS and 18–29 % (21 % on average) to HYRAS. The ensemble spread also covers the observed variability.

Regarding more extreme values, namely the 99.9 % percentile, similar results can be found (Fig. S4 and S5). Again, E–OBS

and HYRAS show a similar behavior for both areas with mean value differences of less than 1 %. The ensemble mean shows a mostly positive bias with deviations of less than 10 % (6 % on average during TP1b) compared to E–OBS for ME* and

6–18 % (average of 10 %) for AL$^*$. Furthermore, there are a distinctly higher spread and variability of the 99.9 % for both, the observations and LAERTES-EU. Except for a few peaks, LAERTES-EU covers the spread of the observations.

## 4.3 Added value of the sample size

In order to demonstrate the added value of the presented LAERTES-EU, we use the signal–to–noise ratio ($S2N$, Eq. A6) for different sample sizes and return periods (cf. Appendix A). Sample size, in this case, means the number of simulation runs. Note that the simulations vary in length (number of years) with a minimum length of 10 years and a maximum of 110 years. In order to reduce the influence of the sample length on the results, the single simulation runs of LAERTES-EU where randomly concatenated using a hundredfold permutation. Observations have a sample size of 1. Again, $S2N$ is calculated for daily spatial

mean precipitation amounts during TP1b only using the HYRAS area.

    For both ME$^*$ and AL$^*$, $S2N$ steadily increases with sample size for all calculated return values meaning a more statistically robust estimate of the return values (Fig. 5). Furthermore, the $S2N$ is lower for higher return periods which is a result of the increasing uncertainty of the best estimate due to less or even no data points for very high return periods. However, $S2N$ also increases with sample size for the very high return periods. The robustness of a 2–year return value estimate of a sample of

size 1 is about the same as the 1000–year estimate for a sample of size 20. This means that even for extremes, which have not been observed yet, some robust statistical analysis can be carried out.

## 5 Long-term variability and trends

The temporal evolution and variability of extreme precipitation throughout the past time period TP1 (1900–2017) and also for the predictions (TP2; 2018–2028) are evaluated in this section. Beside time series of percentiles, we use climate change indices

and statistical distributions. In this section, all land grid cells within the investigation areas ME and AL are used for calculating the daily areal mean precipitation amounts.

## 5.1 Precipitation distributions

Figure 6 shows the evolution of the distribution of areal mean precipitation throughout TP1 and TP2 by treating each decade independently. For the core of the distributions, namely medians, interquartile ranges, and upper whiskers, only small variances

can be found between the different decades which means that there is almost no change for the majority of the precipitation amounts. Nevertheless, a marked positive trend for the uppermost extremes of the distributions appears with maximum values around $18\,\mathrm{mm\,d^{-1}}$ at the beginning of the 20th century and about $24\,\mathrm{mm\,d^{-1}}$ in the 21st century. The distribution for the upcoming decade 2020–2028 shows only small differences to those of the present decade since 2010 with an almost equal median and interquartile range, but slightly higher maximum values (Figure 6, green boxplot). Note that the decade 2010–

2019 contains the years 2018 and 2019 from the predictions, and that the last "decade" 2020–2028 is shorter with 9 years.

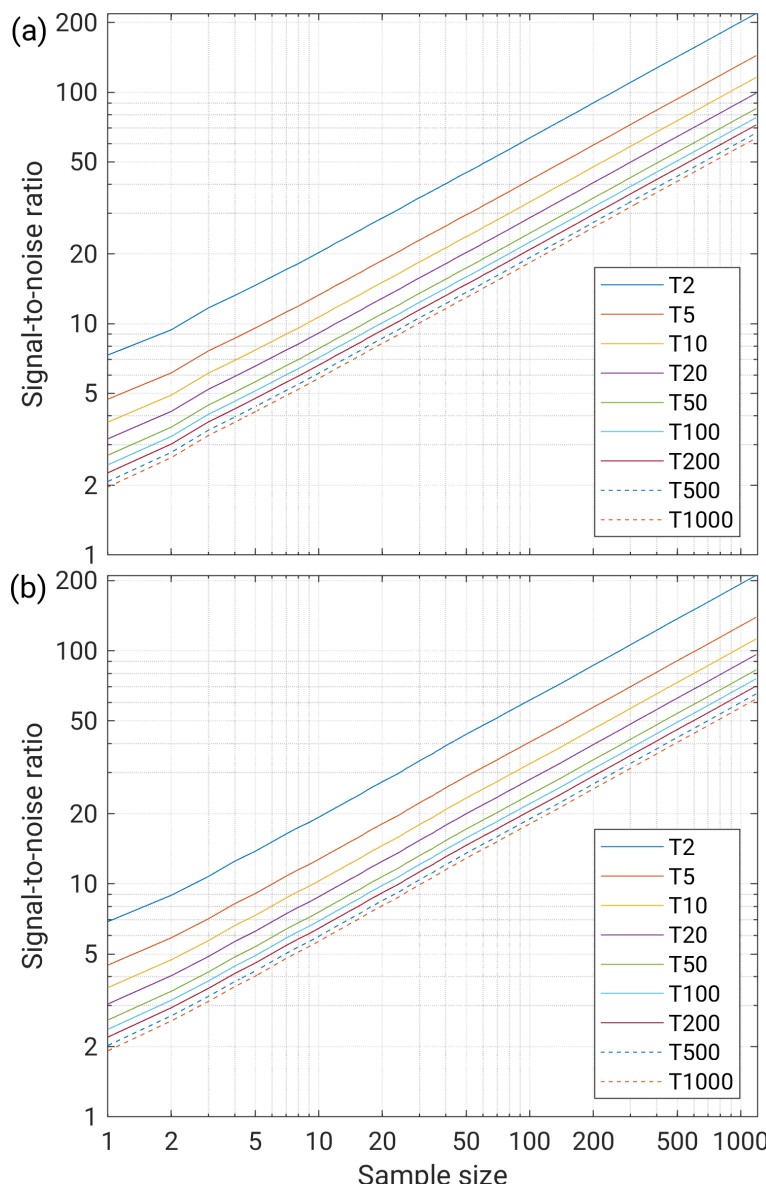

**Figure 5.** Signal–to–noise ratio $S2N$ for different return periods $T$ (colored lines) of daily spatial mean precipitation dependent on the sample size for (a) ME* and (b) AL*. The LAERTES-EU members were randomly stringed together permuting the order a hundred times. The shown $S2N$ is the mean of this permutation.

The boxplot for AL is shown in Fig. S6 and illustrates that not only the high percentiles reveal a decrease in the middle of the century, but the entire distribution is shifted towards lower values. Nevertheless, there is no clear tendency for the maximum values. For the upcoming decade the distribution is similar to that of the present decade in case of median and the upper part of

the distribution (Fig. S6, green boxplot). The interquartile range is reduced due to a increased lower boundary of the boxplot.

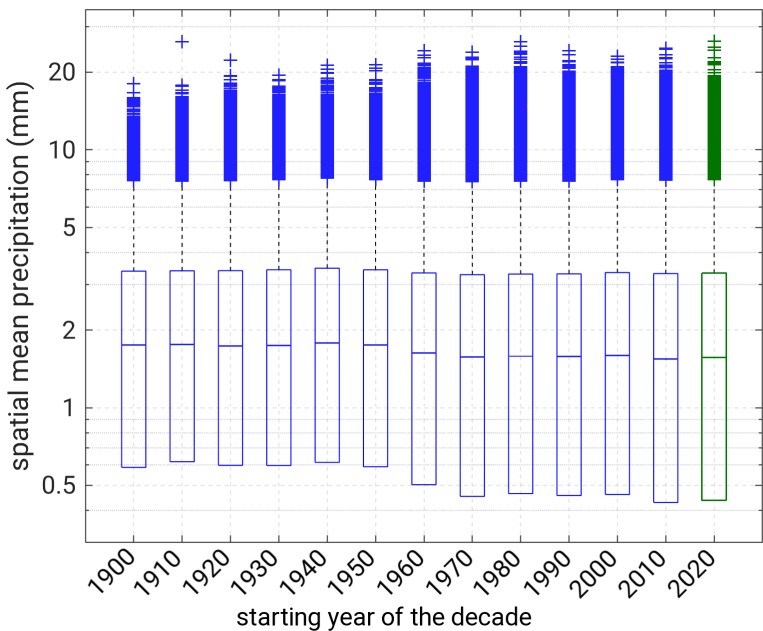

**Figure 6.** Boxplot of the distribution of daily spatial mean precipitation values (including dry days) for ME. Each decade was considered separately. The centerline of a box marks the median; the lower and upper end of the box mark the 25 % and 75 % percentile (interquartile range); the whiskers represent approximately the 99.9 % percentile; the prediction part is marked in green.

## 5.2 Temporal evolution of yearly percentiles

### 5.2.1 Overview

The overall trend during TP1 and TP2 using a linear regression for both areas and percentiles is given in Table 3. While the ensemble mean shows a significant positive trend for ME for both percentiles, a small but significant negative trend can be found for the 99 % of AL, while there is almost no change in the 99.9 % of AL. In all cases, the ensemble spread increases due to both a decrease of the minimum values and an increase of the maximum values both being highly significant. The change of the maximums is stronger than the reduction of the minimums and more pronounced in AL than in ME.

Analogous to Table 3 we analyze the trend for TP1b only (Table S1 in the supplemental material). The tendencies are the same for all cases but less pronounced except for the mean 99.9 % of AL where the negative trend during TP1b is slightly stronger than for the whole time series.

Figure 7 shows the temporal evolution of the 99 % percentile during the 20th and the beginning of the 21st century for the whole LAERTES-EU. As given in Table 3, the lower boundary changes are small, while there is a visible positive trend of the ensemble mean and the upper boundary of the ensemble spread. Note that the larger spread from the 1960s onwards might be artificial due to the decisively larger number of members of data block 4. Nevertheless, there is a clear consistency in the time series for ME.

**Table 3.** Overall trend of daily spatial mean precipitation during TP1 and TP2 (1900–2028) using a linear regression of the yearly series of the 99 % and 99.9 % percentile (pct; wet days only) for ME and AL; Given are absolute values and the relative changes (RC) compared to the climatological mean (climTP; 1961–1990) for the ensemble minimum (min), the ensemble mean, and the ensemble maximum (max) percentile values within LAERTES-EU, and the related significance (p–value; $\alpha = 0.05$).

| area | pct | variable | trend | RC | climTP | $p_\alpha$ |
|------|-----|----------|-------|-----|--------|------------|
|      |     |          | (mm)  | (%) | (mm)   |            |
|      |     | min      | –0.4  | –4.6 | 7.8   | 0.9387     |
| ME   | 99  | mean     | 0.8   | 7.8  | 10.3  | 1.0000     |
|      |     | max      | 2.6   | 19.0 | 13.9  | 1.0000     |
|      |     | min      | –1.0  | –10.9 | 9.0  | 0.9974     |
| ME   | 99.9 | mean    | 1.1   | 8.4  | 13.5  | 1.0000     |
|      |     | max      | 6.7   | 31.0 | 21.6  | 1.0000     |
|      |     | min      | –2.6  | –17.1 | 15.4 | 1.0000     |
| AL   | 99  | mean     | 0.1   | 0.4  | 21.0  | 0.7208     |
|      |     | max      | 5.4   | 18.9 | 28.4  | 1.0000     |
|      |     | min      | –4.3  | –23.9 | 17.8 | 1.0000     |
| AL   | 99.9 | mean    | –0.0  | –0.0 | 27.3  | 0.0000     |
|      |     | max      | 9.0   | 20.0 | 44.7  | 1.0000     |

Some differences emerge for AL (Fig. S7). At first, there is a distinct decrease of the ensemble mean between 1960 and 1970 which might reveal from the rising number of members. As the ensemble matches well with the observations, we presume an overestimation of precipitation in the first half of the 20th century in that region, which could be a result of missing data for the applied dry–day correction. Due to the more complex terrain, the structure of the precipitation fields is more complex, and therefore more sensitive for different types of effects such as the dry–day correction.

The results for the 99.9 % percentile are similar for both areas (Fig. S8 and S9). The positive trend for ME is even more pronounced, while the drop in the 1960s for AL is less visible and therefore, the time series is more constant.

For ME, the evolution of the number of days exceeding the climatological mean percentile reveals a strong positive and significant trend for both the 99 % (Fig. 8, top) and 99.9 % percentile (Fig. S10). The exact values of the climTP mean, the linear regression, the relative change, and the significance can be found in Table 4 (top numbers). For AL, the year–to–year variability is higher and the overall trend is slightly negative (Fig. 8, bottom, and S11) and at least significant for the 99 % percentile. Again, we analyze the trend for TP1b separately (Table 4, bottom numbers). The tendencies for TP1b are the same

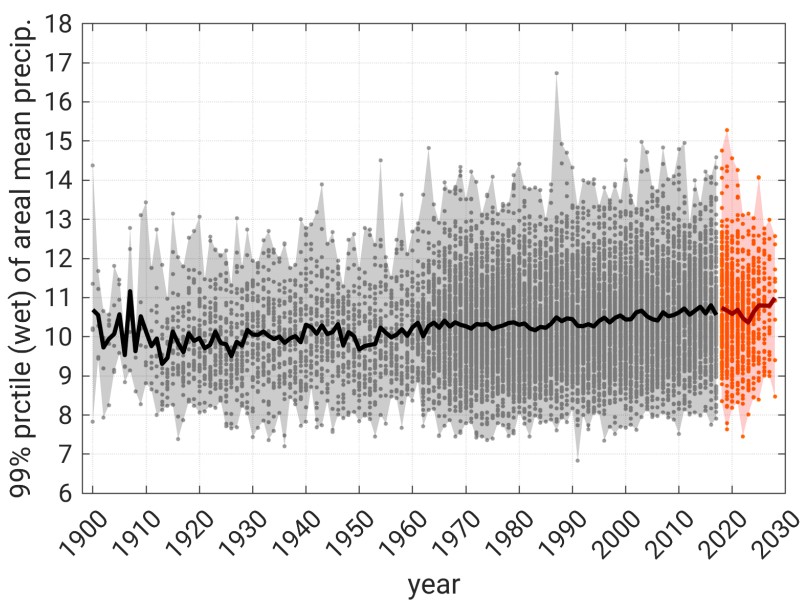

**Figure 7.** Time series of the yearly 99 % percentile of daily spatial mean precipitation (wet days only) for Mid–Europe (ME; land only) of the LAERTES-EU ensemble mean (solid line), and the ensemble spread (minimum to maximum; dots and shaded area) during TP1 (1900–2017; black/gray) and TP2 (2018–2028; reddish).

but less pronounced except for the days exceeding the 99 % percentile in AL, where there is a stronger trend signal in TP1b compared to the whole time series, which is also significant to a high degree.

### 5.2.2 Past trends and periodic oscillations

For a more detailed analysis of trends, the Mann–Kendall test described in Sect. 3.2 is applied to the time series of daily spatial mean precipitation percentiles. Figure 9a shows the relative number of LAERTES-EU members that show a positive or negative trend of the 99 % percentile for ME. Only cases in which more than 60 % of the complete ensemble members reveal the same tendency are then considered for further investigations. For these cases, the ensemble mean trend is calculated (Fig. 9b) and the relative amount of significant members is displayed (Fig. 9c). All cases in which the ensemble reveals ambiguous tendencies are neglected (gray areas).

To a high degree the single members show the same behavior, especially for the longer time series where positive trends are dominant. On a decadal time scale (diagonal line in Fig. 9), some oscillations appear with phases of increasing and decreasing precipitation. This signal might be smoothed as it is not expected that the decadal simulations of data blocks 2 and 4 cover the natural variability at this time scale in detail. Furthermore, these simulations are not expected to be in phase with the long lasting simulations of data blocks 1 and 3. The trends on this time scale reach rates of up to 0.1 mm a$^{-1}$ or 1 mm per decade,

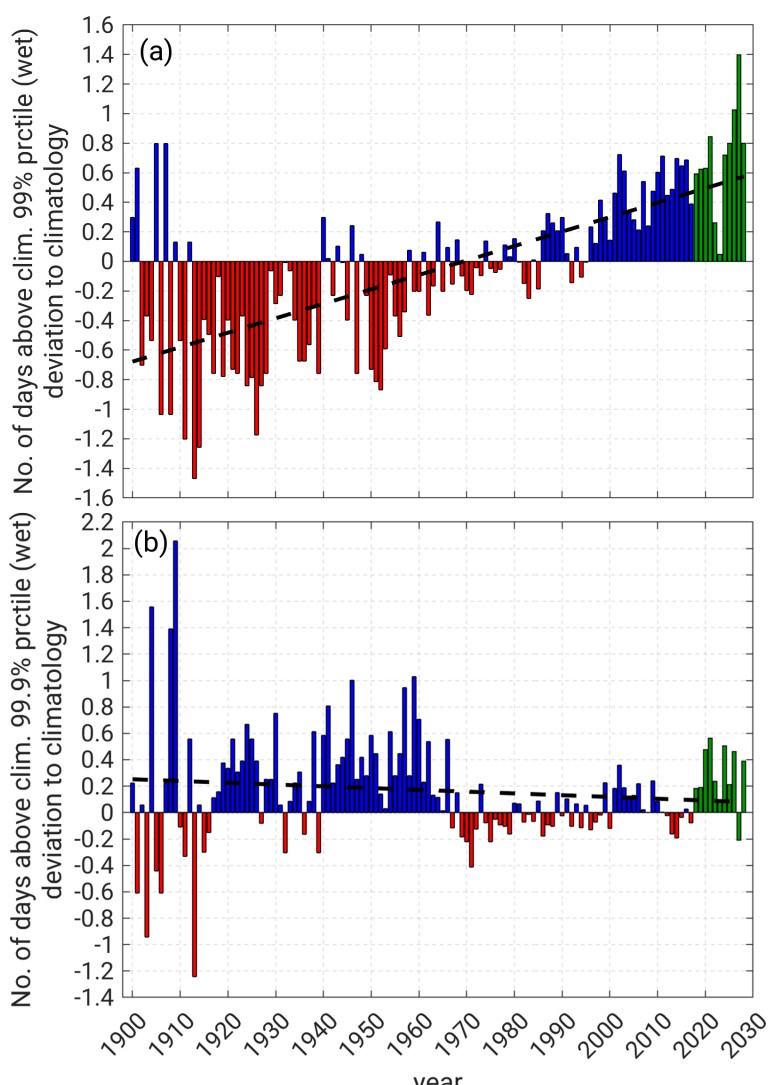

**Figure 8.** Deviation of the LAERTES-EU ensemble mean of the yearly number of days above the 99 % percentile (wet days only) of daily spatial mean precipitation compared to the climatology (climTP; 1961–1990) for (a) Mid-Europe (ME), and (b) the Alps (AL). Red bars indicate negative anomalies (less days), blue bars positive anomalies (more days). The predictions (TP2; 2018–2028) are given in green. The black line indicates a linear regression.

respectively. The overall trend is weaker with rate of 0–0.02 mm a$^{-1}$ or 0–2 mm per century, respectively. Positive trends are more often significant than the negative, while only a small part of the ensemble shows significant trends. Similar results can be found for AL (Fig. S12). The trends on the decadal time scale reach higher rates but the oscillation is less pronounced than in ME. Again, most of the positive trends are significant, while just a few members with negative trends are significant.

**Table 4.** Climatological mean (climTP; 1961–1990) of days per year exceeding the 99 % and 99.9 % percentile (pct; wet days only) for ME and AL, linear regression (LR) and relative change (RC) compared to climTP for different investigation periods (TP), and related significance (p–value; $\alpha = 0.05$).

| area | pct | climTP | TP | LR | RC | $p_\alpha$ |
|------|-----|--------|-----|-------|------|--------|
| ME | 99 | 3.20 | 1+2 | 1.25 | 39 % | 1.0000 |
| | | | 1b | 0.76 | 24 % | 1.0000 |
| | 99.9 | 0.60 | 1+2 | 0.36 | 60 % | 1.0000 |
| | | | 1b | 0.19 | 32 % | 1.0000 |
| AL | 99 | 3.11 | 1+2 | –0.17 | –6 % | 0.8262 |
| | | | 1b | –0.37 | –12 % | 0.9251 |
| | 99.9 | 0.62 | 1+2 | –0.02 | –2 % | 0.7084 |
| | | | 1b | –0.04 | –6 % | 0.2973 |

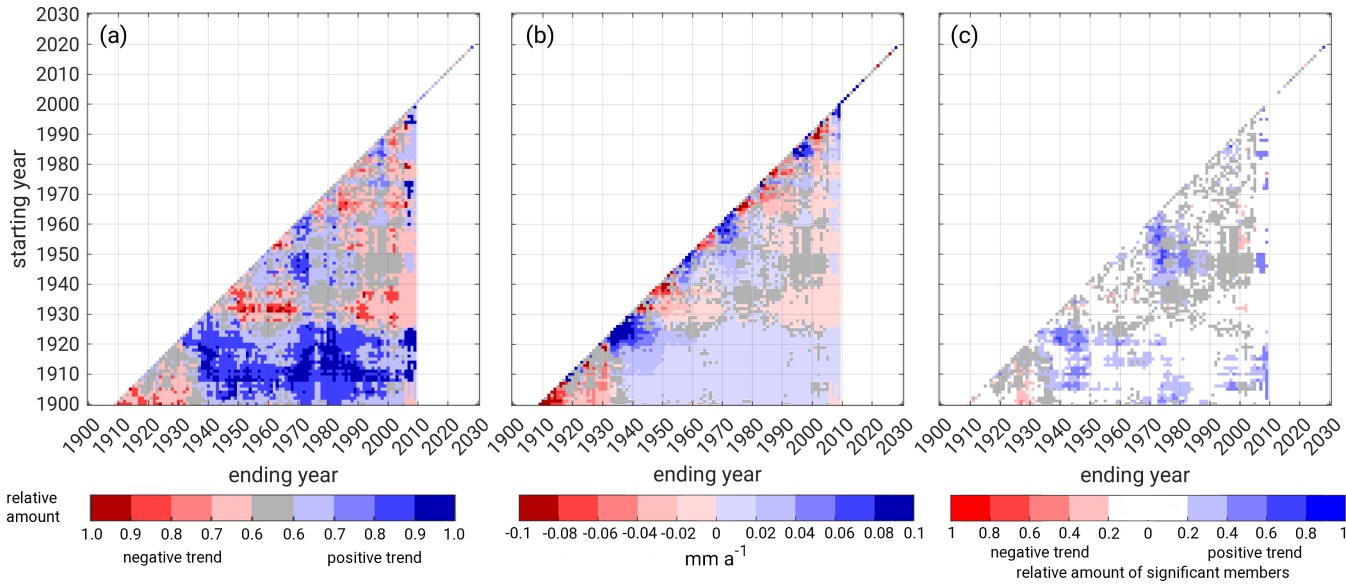

**Figure 9.** Trend analysis of the 99 % percentile (wet days only) of daily spatial mean precipitation for ME with (a) the relative amount of members of LAERTES-EU with a positive (blue) or negative (red) trend; (b) the trend in millimeter per year averaged over the members from (a), and (c) relative amount of members from (a) with a significant trend; cases with no distinct number (less than 60 %) of members with same trend sign are marked in gray in (a)–(c).

For the 99.9 % percentile of ME, large parts of LAERTES-EU show positive trends (Fig. S13). On the decadal time scale a clear sequence of positive and negative trends is visible. Both the increases and decreases are more pronounced than for the 99 % percentile but only a few members are significant. For AL, even more parts of the ensemble have the same tendency of heavy precipitation and a higher number of members have a significant trend (Fig. S14). These trends exceed rates of decisively more than $\pm 0.1$ mm a$^{-1}$. In contrast to the results above, the 99.9 % percentile for AL seems to have a multidecadal oscillation, while the overall trend of the complete time series is negative.

### 5.2.3 Future predictions

With respect for the upcoming decade (TP2; 2018–2028), LAERTES-EU predicts an continuation of the current trend with an increase especially for the 99.9 % percentile (Fig. 7, and S6–S8; reddish area). In comparison to the last decade (2007–2017), the RCM mean of the 99 % percentile increases of about 0.6 % for ME and about 2.1 % for AL. The 99.9 % percentile increases about 2.0 % for ME and 3.0 % for AL.

Further to this absolute change, the number of days exceeding the climatological 99 % percentile shows an increase of 4.9 % for ME and 8.4 % for AL, and 6.7 % (ME) and 22.4 % (AL) in case of the 99.9 % compared to the mean of 2007–2017. This also manifests in the relative anomaly (Fig. 8, and S10–S11; green bars).

Nevertheless, a more detailed trend analysis illustrated in Fig. 9 and also Fig. S12–14 reveals that LAERTES-EU shows no clear tendency for the 99 % during TP2. Just in a few cases, more than 60 % of the members have a similar mainly positive trend signal, which, however, is not significant. In case of the 99.9 % percentile, 60–70 % of the members show a strong positive trend of more than 0.1 mm a$^{-1}$ with 20–40 % of them being significant. Although the tendency for TP2 is ambiguous and less significant, it shows continuity to the present decade.

### 5.3 Climate change indices

The results described in the previous sections also manifest in the considered ETCCDI climate change indices (Table 5). R95pTOT shows a positive trend for ME (Fig. 10a) with a relative change of about 18 % and a strong negative trend of approximately –15 % for AL (Fig. S15). Remarkably, there is a high positive deviation in the first half of the 20th century compared to the climTP amount for AL which might be artificial due to the mentioned problems of the dry–day correction. R99pTOT shows a positive change for ME (Fig. 10b) and a slightly negative trend for AL (Fig. S16). The overestimation for AL in the early century is less pronounced for this index. Considering only the TP1b, the tendencies are the same in all cases. The positive trends for ME are less pronounced, while the negative trends for AL are stronger. The estimated trends are highly significant except for the R99pTOT of AL for the whole time series.

Compared to the present decade, the predictions show a continuation of the positive trend for ME with an increase of 2 % for R95pTOT and 5 % for R99pTOT. In contrast, both indices show a positive trend for AL with an increase of 7 % for R95pTOT and 8 % for R99pTOT, which is a complete reversion of the overall trend.

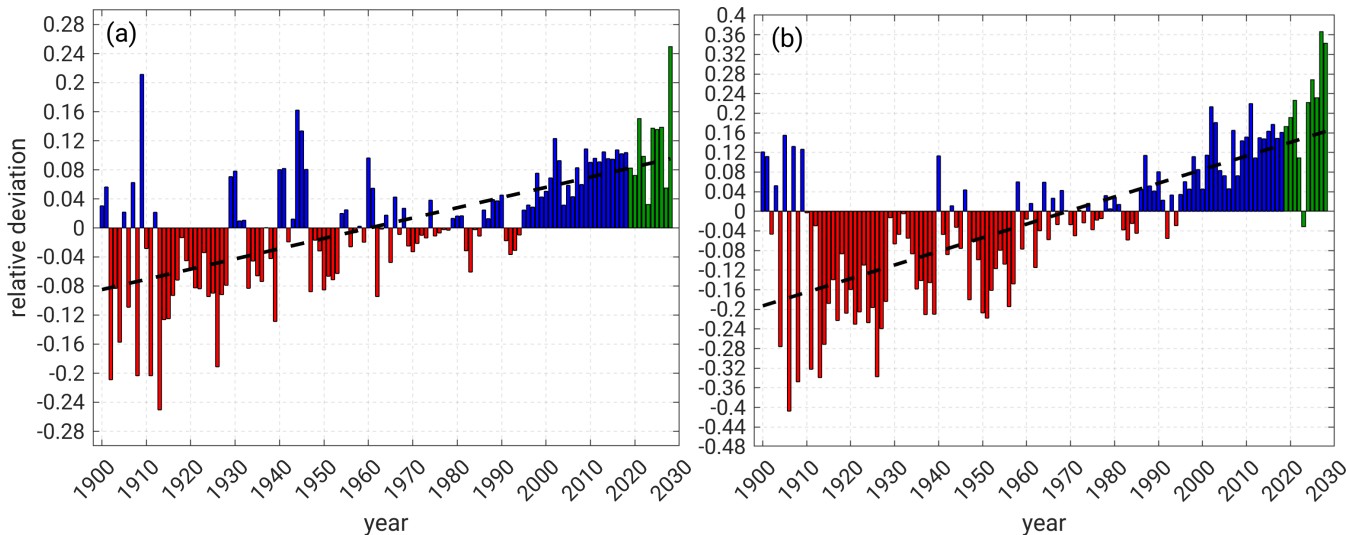

**Figure 10.** Relative deviation of (a) the R95pTOT index and (b) the R99pTOT index of the LAERTES-EU ensemble mean of daily spatial mean precipitation (wet days and land only) compared to the climatology (climTP; 1961–1990; Table 5) for Mid-Europe (ME). Red bars indicate negative (dry) anomalies, blue bars positive (wet) anomalies. The predictions (TP2; 2018–2028) are given in green. The black line indicates a linear regression.

**Table 5.** Climatological mean (climTP; 1961–1990) of ETCCDI quantities for Mid-Europe (ME) and the Alps (AL), linear regression (LR) and relative change (RC) compared to climTP for different investigation periods (TP), and related significance (p–value; $\alpha = 0.05$). Both indices are based on wet days only of daily spatial mean precipitation (land only).

| area | ETCCDI | climTP | TP | LR | RC | $p_\alpha$ |
|------|--------|--------|------|------|-----|-----|
|      |        | (mm)   |      | (mm) | (%) |     |
| ME   | R95pTOT | 157.5 | 1+2  | 28.4 | 18  | 1.0 |
|      |        |        | 1b   | 20.1 | 13  | 1.0 |
|      | R99pTOT | 43.8  | 1+2  | 15.6 | 36  | 1.0 |
|      |        |        | 1b   | 12.2 | 28  | 1.0 |
| AL   | R95pTOT | 306.7 | 1+2  | −46.3 | −15 | 1.0 |
|      |        |        | 1b   | −54.3 | −18 | 1.0 |
|      | R99pTOT | 88.5  | 1+2  | −4.5 | −5  | 0.8953 |
|      |        |        | 1b   | −10.8 | −12 | 0.9891 |

## 6 Summary and Conclusions

We have presented the novel ensemble LAERTES-EU combining various regional climate model simulations done with COSMO–CLM to analyze long-term variability and trends of flood related intensive areal precipitation across central Europe. The whole RCM ensemble was divided into four data blocks depending on forcing data, assimilation schemes, or the initialization of the driving global model MPI–ESM. The setup of the COSMO model remained the same for all simulations. In total, the presented LAERTES-EU consists of over 1100 simulation runs with approximately 12.500 simulated years on a 25 km horizontal resolution.

The focus of investigation was laid on the PRUDENCE regions Mid–Europe (ME) and the Alps (AL). Regarding intensive areal precipitation, we concentrated on high percentiles, namely 99 % and 99.9 % of spatially averaged daily precipitation amounts. Note that it was not expected that LAERTES-EU was able to reproduce historical precipitation events on a daily base in detail, but have a more accurate performance regarding long-term variations, and statistical distributions on a larger scale perspective. Furthermore, the given resolution restricts the consideration of convective processes, so we concentrated on larger scale phenomena.

With respect to our initial research questions, the following main conclusions can be drawn and summed up out of the presented results, which will be discussed more detailed afterwards:

(1) LAERTES-EU is capable of representing the range of extreme areal precipitation similar to the used observational data sets and also fits into the range of previous studies (e.g. Früh et al., 2010). The four data blocks are consistent and have similar precipitation distributions. The ensemble also covers the observed temporal evolution.

(2) The benefits of the large ensemble size manifests in a strong increase of the signal–to–noise ratio beyond the typically used ensemble sizes and in high statistical significances of estimated trends for the ensemble mean. Furthermore, the distribution of precipitation totals is represented in a more concise way taking the limitations of the considered observations into account.

(3) Long-term trends reveal spatial differences in sign and strength. These tendencies are partly significant. Despite a quite large ensemble spread, the ensemble mean shows more explicit results. Distinct oscillations can also be found on shorter time scales (e.g. decades).

(4) The predictions for the upcoming decade show a continuation of past tendencies in terms of both intensity and occurrence frequency for ME without any discontinuity to the previous time period. On the other hand, LAERTES-EU shows no clear signal for AL.

Regarding the validation (1), grid point based intensity–probability–curves (IPCs), areal mean precipitation distributions (internal variability $\sigma_\Gamma$ and linear error in probability space $\overline{L}$), and Q–Q distributions have been analyzed. In all cases, the IPCs of the simulations show an overestimation of precipitation in order of 10–20 % compared to E–OBS. Haylock et al. (2008) found that E–OBS can have a certain negative bias of up to 30 % compared to single ground based punctual observations. Taking

this into account, the IPCs are almost coincident. Furthermore, the IPCs of LAERTES-EU show only small deviation compared
to the HYRAS data set (aggregated to the model grid). The IPCs and also the Q–Q distributions of all four data blocks are
coincident which was a prerequisite for the combination to one large ensemble. The Q–Q distributions of spatially aggregated
mean precipitation reveal less differences between LAERTES-EU and E–OBS, but an underestimation of simulated rainfall
compared to HYRAS by about 10 %. The linear error in probability space $\overline{L}$ shows a good agreement of LAERTES-EU with
observations in terms of the distribution of daily areal mean precipitation totals. For different aggregation intervals from daily
values up to 10-year running means, the internal variability (standard deviation $\sigma_\Gamma$) of LAERTES-EU matches to a high degree
with that of both observations. Note that both quantities $\overline{L}$ and $\sigma_\Gamma$ do not indicate whether the simulated absolute precipitation
values coincide with the observations, but rather show the agreement of statistical properties.

Regarding (2), LAERTES-EU reveals a clear added value due to the large sample size. Estimates of long return periods
are more robust compared to smaller ensembles which is of importance, for instance, for risk and insurance applications.
Furthermore, trends at least in the ensemble mean are highly significant. The IPCs also show a benefit of RCM data compared
to the coarser global model (MPI–ESM) or the 20CR global reanalysis. Regarding extremes, LAERTES-EU includes a broader
range of precipitation totals with even higher values, which are not covered by observations due to their limited temporal
availability. Although the presented results reveal a broad range of realizations within LAERTES-EU, the statistics of the
ensemble mean clearly benefit from the large ensemble size with a better signal–to–noise ratio.

Besides a proper representation of precipitation, long-term trends and temporal variations were of special interest. Regarding
(3), the presented results show a reasonable agreement of LAERTES-EU concerning the temporal evolution of the considered
percentiles of spatially aggregated daily precipitation totals for the different investigation areas. The ensemble spread (min-
imum to maximum) covers the observed variability except a few peaks. The ensemble mean shows a small positive bias
compared to both observational data sets. Throughout the complete time period TP1 (1900–2017), positive and significant
trends can be found for ME in both percentiles (99 % and 99.9 %) and also in the number of days exceeding the climatological
mean (1961–1990). For AL, there is no clear trend signal in the ensemble mean but an increase in the maximum values. In
contrast, the number of days exceeding the climatological mean percentiles is decreasing in this area. Comparing the trends of
TP1 to the shorter TP1b (1951–2006), the tendencies are the same but less pronounced in TP1b. On a decadal time scale, some
oscillations can be found with periods of increasing precipitation and such with decreasing values. Similar results as for time
series of percentiles can be found using climate change indices (ETCCDI).

Regarding (4), the predictions for the next decade 2018–2028 (TP2) reveal ongoing tendencies of heavy precipitation indices.
A special case is AL where the slightly negative trends in the past (TP1) turn to positive ones. Both the continuity for ME and
the reversion for AL appear in all time series, namely the number of days of threshold exceedance, ETCCDI variables, and
investigated percentiles. While there are a clear signal and high significance for the ensemble mean, the trends are ambiguous
and less significant when the ensemble members were considered separately. However, we conclude that this tendencies are
likely as it is a continuation of the results of the present decade. Similar results for parts of LAERTES-EU were found by
Reyers et al. (2019).

Precipitation remains a challenging task for both reanalyses and climate model simulations of the past and the future with partly contrasting results shown by several previous studies. Furthermore, long-term comprehensive observations are not avail-

490 able which makes a validation difficult due to the high spatial variability of precipitation. This also affects analyses of trends or climate variability. What is known is a theoretical increase of the water vapor capacity according to the Clausius–Clapeyron (CC) equation of about 6–7 % per degree of temperature increase (e.g. Trenberth et al., 2003; Berg et al., 2009), which assumes a near constant relative humidity. The CC rate is generally thought to be a proxy for future precipitation projections (Westra et al., 2013). A recent discussion about the validity of the CC rate as an estimate for future projections of heavy precipitation

can be found in Zhang et al. (2017). They pointed out that beside the thermodynamic responses, changes in heavy precipitation may be also influenced by dynamical effects. Furthermore, Pfahl et al. (2017) and Kröner et al. (2017) showed that precipitation trends can be regionally influenced by contributions from both lapse-rate and circulation effects.

The ensemble mean of LAERTES-EU shows an increase of about 1.9 °C for ME and 2.3 °C for AL for the yearly mean 2 m-temperature of spatial means during the 20th century (TP1; 1900–2017). Including the predictions (TP2), the increase is

500 about 2.4 °C for ME and 2.8 °C for AL. For instance, Simmons et al. (2017) found an increase over European land masses of approximately 2 °C in the mean compared to pre-industrial conditions. Moberg et al. (2006) found an increase of about 1 °C for temperature extremes. Thus, LAERTES-EU is within the range of observed changes. The increase in temperature over the entire time period is equivalent to a CC scaling of about 15–20 %. The extracted changes of the high precipitation percentiles for ME make up to 50 % compared to the theoretical CC value. However, the negative tendencies for AL do not fit into this

theoretical estimate.

The presented LAERTES-EU data set can be used for various applications fields. In particular, the simulations are used as input for hydrological modeling and further applications such as flood risk assessments. The presented ensemble in this case can be used as a stochastic weather generator treating the single simulations independently. This leads to the production of a quasi-stochastic hydrological discharge data set. Due to the large ensemble size, estimates of high return periods become more

robust. However, it has to be mentioned that the composition of the four data blocks to one ensemble restricts the temporal homogeneity. Moreover, the validation showed a positive bias of the ensemble mean which, together with the overestimation of low intensities, requires a bias correction to avoid unrealistic discharges. This application as well as the bias correction of LAERTES-EU will be addressed in a consecutive study.

In this study, we have focused on all-year variances, oscillations, or trends. Future investigations can address a seasonal

differentiated analysis of trends and oscillations as well as a more detailed investigation of the spatial distribution of these findings and potential mechanisms behind the observed variability. Previous studies indicated that there is a strong relation between precipitation in Europe and the North Atlantic Oscillation (NAO), especially during wintertime (e.g., Hurrell, 1995; Rîmbu et al., 2002; Haylock and Goodess, 2004; Nissen et al., 2010; Pinto and Raible, 2012). Moreover, Casanueva et al. (2014) found a connection between extreme precipitation and the Atlantic Multidecadal Oscillation (AMO) during the whole

520 year.

*Data availability.* The E–OBS data (Haylock et al., 2008) is online available after registration at https://www.ecad.eu/download/ensembles/ ensembles.php. The 20CR data (Compo et al., 2011) can be found on https://www.esrl.noaa.gov/psd/data/20thC_Rean/. HYRAS (Rauthe et al., 2013) can be requested at the German Weather Service (DWD). The RCM data (MiKlip data) will be made available via the CERA database (http://cera-www.dkrz.de/; last access: July 2019) of the German Climate Computing Center (DKRZ).

 **Appendix A: Statistical Quantities**

The linear error in probability space $L$ uses the difference of probabilities $\Delta C$ defined as:

$$\Delta C_{\mathrm{r}}(x) = ecdf_{mod,r}(x) - ecdf_{obs}(x) \,, \tag{A1}$$

where $ecdf_{mod,r}$ is the empirical cumulative density function of the model run $r$, and $ecdf_{obs}$ that of the observation up to precipitation intensity $x$. The linear error in probability space $L_r$ for a model run $r$ is then defined as (Déqué, 2012; Wahl et al., 2017):

$$L_{\mathrm{r}} = \frac{1}{n} \cdot \sum_{x=1}^{n} |\Delta C_{\mathrm{r}}(x)| \,. \tag{A2}$$

$L_r$ describes the mean value of $\Delta C_r$ over the entire range of precipitation intensities $x$ grouped into $n$ classes. Using absolute values avoids a compensation of positive and negative values. The better both distributions coincide, the lower the value of $L_r$. The ensemble mean of $L_r$ is given by:

$$\overline{L} = \frac{1}{M} \sum_{r=1}^{M} L_{\mathrm{r}} \,, \tag{A3}$$

with $M$ being the total number of simulation runs.

The model performance on different frequency intervals is further validated using the standard deviation of a gamma distribution $\sigma_\Gamma$ (Wilks, 2006), which is given by:

$$\sigma_\Gamma^2 = \alpha \beta^2 \,. \tag{A4}$$

In this formulation, $\alpha$ is the shape parameter of the gamma distribution, and $\beta$ its scale parameter.

The quantile–quantile analysis uses the Pearson correlation coefficient (Wilks, 2006) given by:

$$R = \frac{\sum\limits_{k=1}^{N} \{[y_k - \overline{y_k}] \cdot [x_k - \overline{x_k}]\}}{\sqrt{\sum\limits_{k=1}^{N} [x_k - \overline{x_k}]^2} \cdot \sqrt{\sum\limits_{k=1}^{N} [y_k - \overline{y_k}]^2}} \,, \tag{A5}$$

with the data series $x$ and $y$ of length $N$. The range of $R$ is $R \in [-1; +1]$ with a perfect anti-correlation at $R = -1$ and a perfect correlation at $R = +1$.

The signal–to–noise ration $S2N$ in this study is defined as:

$$S2N = \frac{RV_{\mathrm{T,Gumbel}}}{CI_{90,\mathrm{T}}} \,, \tag{A6}$$

with the return level $RV$ of the Gumbel distribution at return period $T$ divided by its $90\,\%$ confidence interval at $T$ (Früh et al., 2010). Small values of $S2N$ indicate a more uncertain estimate, high values a more robust one. The Gumbel distribution

(Wilks, 2006) is an extreme value type-I distribution and often used for return period estimation. Its cumulative density function (cdf) is given by:

$$F(x) = \exp\left(-\exp\left(-\frac{x-\beta}{\alpha}\right)\right) , \tag{A7}$$

with the free parameters $\beta = \sigma\sqrt{6} \cdot \pi^{-1}$ and $\alpha = \bar{x} - \gamma\beta$, where $\sigma$ is the standard deviation of the sample $x$ assuming a normal distribution, and $\gamma = 0.57721$ Euler's constant. For $x$, usually a series of yearly maximum values is used. The relationship between the cdf and the return period $T$ is given by (Wilks, 2006):

$$T = \frac{1}{1-F(x)} . \tag{A8}$$

## Appendix B: ETCCDI quantities

Two out of the 27 indices introduced and recommended by the Expert Team on Climate Change Detection and Indices[4] (ETCCDI; Karl et al., 1999; Peterson, 2005) are used in this study. R95pTOT describes the annual total precipitation sum of all values above the climatological 95 % percentile of wet days ($RR > 1\,\mathrm{mm}$) during the reference period 1961–1990. The R95pTOT of the year $k$ is defined as:

$$\mathrm{R95pTOT_k} = \sum_{w=1}^{W} RR_{wk} \quad \forall\, RR_{wk} > RR_{p95} , \tag{B1}$$

where $RR_{wk}$ is the daily precipitation amount on a wet day during year $k$, $RR_{p95}$ is the climatological 95 % percentile, and $W$ the total number of wet days in year $k$. Analogously, the R99pTOT is defined replacing the 95 % with the 99 % percentile:

$$\mathrm{R99pTOT_k} = \sum_{w=1}^{W} RR_{wk} \quad \forall\, RR_{wk} > RR_{p99} . \tag{B2}$$

## Appendix C: Trends and Significance

A Mann–Kendall Test (Mann, 1945; Kendall, 1955) is performed for the detection of trends and its related significance. To account for possible oscillations within long time series, we first split the complete time series into sub-series with a minimum length of 10 years and up to over 100 years (trend matrix). The Mann-Kendall Test uses a standardized test statistic $S_\tau$ following a standard Gaussian distribution (SGD). $S_\tau$ is given by:

$$S_\tau = \begin{cases} \frac{\tau-1}{\sqrt{\sigma_\tau^2}} & , \tau > 0 , \\ 0 & , \tau = 0 , \\ \frac{\tau+1}{\sqrt{\sigma_\tau^2}} & , \tau < 0 . \end{cases} \tag{C1}$$

---

[4]http://etccdi.pacificclimate.org/

Here, $\tau$ is known as the Kendall's $\tau$ and $\sigma_\tau^2$ is the variance of the standard Gaussian distribution (SGD). A detected trend is significant if $S_\tau$ lies within the upper and lower quantile $z$ of the SGD at a given significance level $\alpha$ with $S_\tau \in \left[ z_{\frac{\alpha}{2}} \sigma_\tau ; z_{1-\frac{\alpha}{2}} \sigma_\tau \right]$, respectively (Yue et al., 2002).

Yue et al. (2002) pointed out some weaknesses of the Mann–Kendall test in case of inherent autocorrelation. To avoid a distortion of the statistic by autocorrelation, Yue et al. (2002) presented the Trend–Free Pre–Whitening (TFPW) method. The first step is the estimation of a linear trend between two time steps $t = i$ and $t = j$ using the Theil-Sen Approach (TSA; Theil, 1950; Sen, 1968). The slope $b$ of this linear regression is given by:

$$b = median \left( \frac{x_j - x_i}{j - i} \right) , \forall i < j . \tag{C2}$$

In a second step, the original time series $x$ is detrended by subtracting $b$ at each time step $t$:

$$x_t' = x_t - b \cdot t . \tag{C3}$$

Afterwards, the lag-1 autocorrelation coefficient $r_1$ is removed from the trend-free series $x'$:

$$x_t'' = x_t' - r_1 \cdot x_{t-1}' , \tag{C4}$$

where $r_1$ is given by:

$$r_1 = \frac{\frac{1}{N-1} \cdot \sum_{i=1}^{N-1} \left( x_i' - \overline{x'} \right) \cdot \left( x_{i+1}' - \overline{x'} \right)}{\frac{1}{N} \cdot \sum_{i=1}^{N} \left( x_i' - \overline{x'} \right)^2} . \tag{C5}$$

The modified TFPW time series $x^*$ results by re-adding the TSA-slope $b$:

$$x_t^* = x_t'' + b \cdot t . \tag{C6}$$

This modified time series conserves the trend, but is free of autocorrelation. The Mann–Kendall Test is performed on the TFPW time series $x^*$. According to Yue et al. (2002), TFPW has to be considered in cases with non-zero TSA-slope and significant lag-1 autocorrelation. The significance of a trend or autocorrelation is tested on the 90 % ($\alpha = 0.1$), 95 % ($\alpha = 0.05$), and 99 % ($\alpha = 0.01$) significance level.

*Author contributions.* FE, LAK, HF, and JGP designed the study. HF performed (parts of) the RCM simulations. LAK applied the dry–day correction. FE did the analysis and plots, and wrote the initial draft. All Authors contributed with discussions and revisions.

*Competing interests.* The authors declare that they have no conflict of interest.

*Acknowledgements.* The authors thank the National Centers for Environmental Prediction (NCEP) for providing the 20CR data. We acknowledge the E–OBS data set from the EU-FP6 project ENSEMBLES (http://ensembles-eu.metoffice.com) and the data providers in the ECA&D project (http://www.ecad.eu). We also thank the German Weather Service (DWD) for providing HYRAS. In addition, we thank the German Climate Computing Center (DKRZ, Hamburg) for computing and storage resources. We thank the BMBF for funding the MiKlip project (contract number 01LP1129A/D) and AON for funding the extreme weather events project. We acknowledge open access publishing fund of Karlsruhe Institute of Technology (KIT). JGP thanks the AXA Research Fund for support. We thank the reviewers for their valuable comments that helped to improve this study, and the handling editor for guidance throughout the entire process.

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
