# Peer review of "Long-term Variances of Heavy Precipitation across Central Europe using a Large Ensemble of Regional Climate Model Simulations"

_Earth System Dynamics, 2019_

## Referee Comment (RC1) · Anonymous Referee #1 · 10 Oct 2019

This manuscript addresses the issue of heavy precipitation in RCM simulations. This is a very timely issue with importance for many sciences, which rely on RCM simulations. My fundamental concern with this study is the first conclusion ("Extreme precipitation is well represented in LAERTES-EU."). The same is expressed in the authors' short summary ("The simulations show a good agreement with observations for both statistical distributions and time series of heavy precipitation."). I am sorry, but I just can not see enough support for this crucial statement in the manuscript.

1) The authors state that E-OBS underestimates precip by almost a third. To me, this means that these data are not useful to evaluate the performance of extreme value

simulations. As E-OBS is only available for land surfaces, I also find it surprising that the ME box includes parts of the North Sea.

2) The evaluation using IPCs is good, but doing this on a highly aggregated level seems to limit the opportunity to really test the simulated precip. Here I would like to see more creative tests such as IPCs for smaller areas and/or IPCs for certain seasons. As the analysis is done now, there is a risk for error compensation.

3) I am sorry, but I do not see how the Q-Q plots help to evaluate the performance for heavy precip. If anything, the total precip is evaluated. But even then, comparing cumulative values introduces a spurious correlation, and on top of that, $R2$ is no suitable measure as a value of one does not ensure a 'perfect' model. May be I miss something here, but I find this analysis not convincing.

4) The 99% of precip (=around 10 mm) is not really 'heavy precipitation.'

Minor comments:

P4L115: "more or less independent simulations". This needs to be clarified. In some respect, these simulations might be independent, but as the same RCM is used, the simulations obviously are dependent!

P6L149: does this mean there was a bias correction? Were extreme precipitation simulations affected by this at all? I assume not but would like to get some clarification.

P7L197ff: I can see the argument that GCMs underestimate heavy precip, but the same argument should, although to a smaller degree, apply to RCMs. So, what is the physical reason that RCMs 'tend to overestimate precipitation intensities'?

---

## Referee Comment (RC2) · Anonymous Referee #2 · 26 Oct 2019

This paper analyzes long-term trends of heavy precipitation in multiple dynamically downscaled simulations for the historical period and the near future over Europe. The different sets of simulations are validated against gridded observations and tested whether they can be combined to a large ensemble for the detection of trends in the historic time period. This paper is relevant in terms of assessing the possibility of combining various simulations from the same RCM with varying driving data. As well as, the detection of trends within the historic time period.

General comments:

1) Please be more clear about what you are showing in the figures. In most cases it

wasn't clear to me if you are showing the ensemble mean or a metric with pooled data from the entire ensemble.

2) Several sections need more clarification on what was analyzed and for what spatial extend and aggregation.

Major comments:

. . . Model evaluation:

1) I have concerns with the comparison of E-OBS, CCLM and HYRAS over the sub-region AL. It is not clear to me whether the comparison was only performed for the HYRAS grid cells, which cover a substantially smaller area than E-OBS and CCLM, or whether E-OBS and CCLM represent the entire AL domain compared to a much smaller area in HYRAS. On P8-L204f you state this concern yourself '[. . .] which might be a reason for the vanished differences between E-OBS and HYRAS and the resulting specious deviations to the RCM'. Did you compare the three datasets for the HYRAS grid cells only? If not please do so.

2) Further, you state that '[. . .] by taking into account all grid points and all time steps within the investigation are (P6-L151)', does this mean that for both ME and AL you have included ocean grid cells in the spatial average of the RCM data? For both do-mains gridded observational datasets are only available over land. Please clarify this, and in any case ocean grid cells were included remove them from the comparison.

2) In P8-L212f you state that '[. . .] HYRAS was aggregated to the E-OBS/RCM grid [. . .]'. However, you first mentioned this here for the Q-Q plots, so can I assume that the IPC's in Figure 2 are also based on aggregated HYRAS data? Please clarify this and if the aggregation of HYRAS applies to all related analysis then please move this detail to the methods section.

3) Further, if I understand correctly the evaluation is based on the TP1b time period (1950-2017), however the HYRAS data is only available for the period 1951-2006.

Please comment on why the analysis wasn't based on the shorter HYRAS time period. I would recommend doing the analysis for 1951-2006.

4) The evaluation on such a highly aggregated level poses a risk of error compensation. It might be better to do the evaluation for each grid cell first (e.g. calculating the RMSE) and afterwards averaging the error metric.

5) Based on the concerns above, I don't really agree with your first point in the conclusion 'Extreme precipitation is well represented in the LAERTES-EU [P20-L352]'.

. . . Added-Value:

6) Regarding your conclusions on the IPCs showing '[. . .] a clear added value of RCM data compared to coarser global models'. From that one figure I don't really see the added value, since you haven't compared the driving GCM with RCM simulation. You have compared the IPC's to the 20CR reanalysis dataset. Because of the spatial averaging over such a large area, it might be that the trends in the GCM and RCM might be very close to each other.

. . . CC-scaling

7) Your conclusion on the trends following the CC-scaling in your conclusion [P21-L379ff] are flawed. If you make a statement like this, please perform the temperature scaling with the LAERTES-EU temperature data and not by relating the precipitation change to a temperature approximation from another study. Please see Kröner et al (2017) and Pfahl et al (2017) for other effects than thermodynamics. Kröner et al (2017), Climate Dynamics, https://doi.org/10.1007/s00382-016-3276-3 Pfahl et al (2017), Nature Climate Change, https://doi.org/10.1038/nclimate3287

Minor comments:

P2-L25f: see also Zhang et al (2017) for a discussion on CC scaling Zhang et al (2017), Nature Geosciences, DOI: 10.1038/NGEO2911

P2-L27-34: Please add some more recent literature on this topic, e.g. Fischer and Knutti (2016), Nature, DOI: 10.1038/NCLIMATE3110 Alexander (2016), Weather and Climate Extremes, http://dx.doi.org/10.1016/j.wace.2015.10.007 Barbero et al (2017), GRL, doi:10.1002/2016GL071917

P2-L45f: Please add a view more recent studies on trends in European floods. E.g.: Blöschl et al (2017), Science, DOI: 10.1126/science.aan2506; Blöschl et al (2019), Nature, https://doi.org/10.1038/s41586-019-1495-6

P2-L49f: Connection of Heavy Precipitation over central Europe and cyclones, see also Hoffstätter et al (2017), Int. Journal of Climatology, https://doi.org/10.1002/joc.5386

P2-L55f: Also see van der Wiel et al (2019) and Martel et al (2019) for the added value of large ensembles for flood risk or return periods of heavy precipitation van der Wiel et al (2019), GRL, https://doi.org/10.1029/2019GL081967 Martel et al (2019), Journal of Climate, https://doi.org/10.1175/JCLI-D-18-0764.1

P4-L114f: Please elaborate more on what you mean by '[. . .] more or less independent simulations'

P5-L139: What do you mean by un-initialized? Please clarify this for the reader, that by initialized you mean initialized by observational(-like) salinity and other variables, whereas the un-initialized data originate from a normal CMIP5 historical simulation. I had to go to Marotzke et al (2016) to understand what was meant by this.

P6-L157f: Are the 99th and 99.9th percentile based on all days or wet-days only? If you want to look at heavy precipitation it might be better to look at wet days only. Like this the values would not be affected by the dry-day adjustment as much. Further, it is not clear to me if you have first spatially aggregated and then calculated the percentiles, or the other way around. Please comment on whether you think that this will make a difference to your results. This could maybe also solve your concerns on P15-L282f '[. . .] an overestimation of precipitation [. . .], could be a result of missing data for the

applied dry-day correction.'

P7-L189: Could you briefly comment on why you chose the old 1961-1990 period as your reference climatology.

P9-L218f: Your conclusion to Table 2 stating that there is a higher correlation when driven by MPI-ESM-HR versus lower resolution MPI-ESM-LR is technically correct, however the differences are so marginal that I find it difficult to attribute the differences to resolution of driving data. Especially, when not only the resolution is different in the HR and LR simulations, but also the initialization. Maybe add a short sentence ['However, differences are only marginal.'].

Chapter 4.3: This is a nice analysis that shows the benefit of large ensembles, however since you are not looking at return values afterwards it could be nice to highlight another strength of large ensemble namely isolating the forced response from internal variability. Since you are looking at trends and variability this could be a better fit. But this is just a suggestion to improve the flow of the paper. Like a said it is a nice analysis as is.

Figure 8a: Shouldn't there be also some more positive anomalies in the climTP period? Did I miss something? Because if you base the annual anomalies on this period, shouldn't you be having positive and negative anomalies within this period?

Figure 9: Nice plots!

P22-L396f: '[. . .] can be used as input for hydrological modeling'. In general, yes and especially when looking at higher return levels of floods. However, as mentioned a few lines above this ensemble is restricted by temporal homogeneity, which can play a very important role in hydrology.

Technical Corrections:

Table 1: Projections for the period 2020-28 are missing

[Figure]

Figure 2, 3: Please add the years of the period ([. . .] TP1b (1950-2017)). I had to go back and look for the TP1b definition. But I would anyway suggest changing the period to 1951-2006 (see major comments).

P3-L69: Typo ('Regionla', 'Regional')

P5-L128: grammar (replace 'it' with 'the')

Figure 4: Please clarify what the RCM spread is. I assume it to be Min-May, right?

P20-L342: grammar ('estimate' instead of 'estimated')

---

## Author Response (AR1)

**List of Major Changes**

- 1) For the validation of LAERTES (Sect. 4), we changed:
  - (a) the time period TP1b (old: 1950-2017) to the HYRAS period (new: 1951-2006)
  - (b) the investigation areas ME and AL are reduced to the corresponding HYRAS grid cells, indicated by ME\* and AL\*
  - (c) the estimation of the percentiles is newly done considering only wet days with R>0.1mm
  - (d) we included additional quantities like the linear error in probability space L and a frequency analysis
  - (e) Figure 5 was re-done using perturbed sequences of concatenated ensemble members to reduce the influence of member size on the shape of the curves.
- 2) Figure 8a was wrong (old version), we replaced it with the correct one
- 3) More precise captions of figures and tables in both the manuscript and the supplemental material
- 4) We rearranged the Section 2.2 describing the ensemble data to be more precise and consistent
- 5) We rearranged the introduction
- 6) We adjusted the conclusions accordingly

**Point-by-point response to Reviewer #1**

Florian Ehmele on behalf of all co-authors October 31, 2019 Update: March 9, 2020

Dear reviewer No. 1,

Thank you very much for your work and the useful and valuable comments that will help to improve the scientific quality of our manuscript. Below you will find your comments given in gray and our responses to the individual points in black. Please also consider our comments to Reviewer 2 as there is some coincidence of the comments and the corresponding answers.

This manuscript addresses the issue of heavy precipitation in RCM simulations. This is a very timely issue with importance for many sciences, which rely on RCM simulations. My fundamental concern with this study is the first conclusion ("Extreme precipitation is well represented in LAERTES-EU."). The same is expressed in the authors' short summary ("The simulations show a good agreement with observations for both statistical distributions and time series of heavy precipitation."). I am sorry, but I just can not see enough support for this crucial statement in the manuscript.

You and also RC2 have the same concerns about this first conclusion. Thinking about this a second time we came along that this statement might be too general. It was meant that heavy precipitation is consistent in all parts of LAERTES-EU and that our results fit in the range of previous studies (e.g. Früh et al., 2010) and also in the range of observations knowing that the used observational data sets have uncertainties as well. We will rewrite this to be more precise what was meant to be stated here. However, we do think that, for instance, the IPCs do support the statement in terms of the statistical distribution of precipitation values. The time series of LAERTES-EU (ensemble mean) is within the range of both analyzed observational data sets, and the ensemble spread covers the observed variability. Please note that it was never intended that LAERTES-EU shows a one-by-one agreement with historical events.

1) The authors state that E-OBS underestimates precip by almost a third. To me, this means that these data are not useful to evaluate the performance of extreme value simulations. As E-OBS is only available for land surfaces, I also find it surprising that the ME box includes parts of the North Sea.

E-OBS has some limitations, like a certain underestimation especially for extremes, but these have been mentioned by several previous studies like Haylock et al (2008) or Hofstra et al. (2009), and mainly appear in a grid point comparison with measurement sites. As these and other studies already used and analyzed E-OBS, we follow their conclusions and did not perform a further analysis on data quality. Keeping the limitations in mind, E-OBS can be useful for evaluation. Unfortunately, there is no other high-resolution daily precipitation data set available that covers entire Europe for a quit long time period. As the focus of this study is on intensive areal precipitation, we think it does not make sense to use single ground based observations that potentially are available for longer time scales, or in terms of the focus on longterm evolution other products like satellite data with a very limited time frame are not helpful and also have limitations. We will add a comment on this situation to the revised manuscript.

The prudence region are defined as regular lat/lon-boxes and therefore cover ocean areas as well. But, in every case ocean grid cells have been set to a missing value in every dataset and therefore, they are not in the results. We will add a sentence on that in the method section for clarification.

2) The evaluation using IPCs is good, but doing this on a highly aggregated level seems to limit the opportunity to really test the simulated precip. Here I would like to see more creative tests such as IPCs for smaller areas and/or IPCs for certain seasons. As the analysis is done now, there is a risk for error compensation.

For the IPCs no aggregation was done. We take all grid point values within the investigation area (e.g. ME) and at all timesteps into account and group them into a histogram giving the probability of occurrence (=IPC). On purpose, we only use all year data and no seasonal differences as the paper would have become to long doing a seasonal analysis for every part of it. This study was meant to be an introduction to LAERTES-EU and some long term investigations on the upper part of the precipitation distribution.

For a more appropriate evaluation of LAERTES-EU, we will follow Reviewer 2 and include some further analysis using other methods and quantities. In particular, we will add a frequency analysis.

3) I am sorry, but I do not see how the Q-Q plots help to evaluate the performance for heavy precip. If anything, the total precip is evaluated. But even then, comparing cumulative values introduces a spurious correlation, and on top of that, R2 is no suitable measure as a value of one does not ensure a 'perfect' model. May be I miss something here, but I find this analysis not convincing.

We will restructure the evaluation part and will introduce additional skill measures in the evaluation (see also answers to RC2), for example, the linear error in probability space L (e.g. Potts et al., 1996)

**4) The 99% of precip (=around 10 mm) is not really 'heavy precipitation.'**

Technically, this is correct. However, the focus of this study is intensive areal precipitation, which is related to widespread flooding along the great major river networks. For a single grid cell, 10 mm is no big deal but 10 mm on average over a large area such as the Rhine catchment or the entire PRUDENCE region is quite a lot. Maybe the term 'heavy precipitation' somehow is irritating at this point. We will include a clarification on that at the beginning of the revised manuscript. Furthermore, please see also the report of Reviewer 2. We will change the percentile calculation to wet days only, as currently dry spells are included.

**Minor comments:**

P4L115: "more or less independent simulations". This needs to be clarified. In some respect, these simulations might be independent, but as the same RCM is used, the simulations obviously are dependent!

We agree that this formulation is inept and we will remove it. What was meant is that the temporal evolution of the day-to-day weather in hindcasts is independent after a few weeks. The statement did not refer to the model system. The ensemble does not cover the full range of uncertainty, namely the model uncertainty. But, in the context of the paper we regard this as an advantage, since the data set is homogeneous over time due to the consistent model setup.

P6L149: does this mean there was a bias correction? Were extreme precipitation simulations affected by this at all? I assume not but would like to get some clarification.

The dry-day adjustment only corrects the number of days without precipitation (R<0.1mm/day) in the model as RCMs tend to produce too much days with very small rainfall amounts (drizzle effect; Berg et al., 2012). The absolute values (R>=0.1mm) are not affected. A bias correction, meaning an adjustment of the absolute precipitation values by e.g. a quantile mapping, was not performed at this stage. In a consecutive study (Kautz et al., planned submission in summer 2020), a specific application of LAERTES-EU for hydrological issues will be presented for which such a bias correction is mandatory. For any other application, a reduction of the drizzle effect has to be done anyway.

P7L197ff: I can see the argument that GCMs underestimate heavy precip, but the same argument should, although to a smaller degree, apply to RCMs. So, what is the physical reason that RCMs 'tend to overestimate precipitation intensities'?

Two effects are of relevance at this point and which act together. The limited time period of observations results in unknown distributions, especially at the heavy tail. In a dataset of 65 years, extreme events with return periods of 100 years or more are not represented in a statistically robust way. The RCM has a physical background when calculating precipitation amounts which makes it possible to reach higher than observed values. Furthermore, the huge number of simulations allows for a more robust estimate of the high-intensity tail of the distribution, whereas the observations display only a few single events in this range.

**Point-by-point response to Reviewer #2**

Florian Ehmele on behalf of all co-authors October 31, 2019 Update: March 9, 2020

Dear reviewer No. 2,

Thank you very much for your work and the useful and valuable comments that will help to improve the scientific quality of our manuscript. Especially your suggestion on how to implement the comments to the paper are very useful. Below you will find your comments given in gray and our responses to the individual points in black. Please also consider our comments to Reviewer 1 as there is some coincidence of the comments and the corresponding answers.

This paper analyzes long-term trends of heavy precipitation in multiple dynamically downscaled simulations for the historical period and the near future over Europe. The different sets of simulations are validated against gridded observations and tested whether they can be combined to a large ensemble for the detection of trends in the historic time period. This paper is relevant in terms of assessing the possibility of combining various simulations from the same RCM with varying driving data. As well as, the detection of trends within the historic time period.

General comments:

1) Please be more clear about what you are showing in the figures. In most cases it wasn't clear to me if you are showing the ensemble mean or a metric with pooled data from the entire ensemble.

Thank you for this feedback. We will change the figure caption to be more precise and accurate to become more clear. Therefore we will also include the related minor comments you wrote below.

2) Several sections need more clarification on what was analyzed and for what spatial extend and aggregation.

Going through your major and minor comments below and include them into the new version of the manuscript, we think this will clarify a lot of points within the text. Please see our detailed comments to the specific points below.

Major comments:

**... Model evaluation:**

1) I have concerns with the comparison of E-OBS, CCLM and HYRAS over the sub-region AL. It is not clear to me whether the comparison was only performed for the HYRAS grid cells, which cover a substantially smaller area than E-OBS and CCLM, or whether E-OBS and CCLM represent the entire AL domain compared to a much smaller area in HYRAS. On P8-L204f you state this concern yourself '[...] which might be a reason for the vanished differences between E-OBS and HYRAS and the resulting specious deviations to the RCM'. Did you compare the three datasets for the HYRAS grid cells only? If not please do so.

This is a crucial comment, thank you for that. Checking our data we found out that the analysis for the AL region indeed was performed for the entire region for CCLM and E-OBS but HYRAS only for available grid cells. We will fix this in the new manuscript

**version and also double check our results for the ME region to be done on HYRAS grid cells only. We will name the sub-areas of ME and AL, in which HYRAS is available, with an asterisk (ME\* or AL\*), to be clear.**

2) Further, you state that '[...] by taking into account all grid points and all time steps within the investigation are (P6-L151)', does this mean that for both ME and AL you have included ocean grid cells in the spatial average of the RCM data? For both domains gridded observational datasets are only available over land. Please clarify this, and in any case ocean grid cells were included remove them from the comparison.

In every case, ocean grid cells have been set to a missing value in every data set and therefore, they are not in the results. We will add a sentence on that in the method section for clarification.

2) In P8-L212f you state that '[. . .] HYRAS was aggregated to the E-OBS/RCM grid [. . .]'. However, you first mentioned this here for the Q-Q plots, so can I assume that the IPC's in Figure 2 are also based on aggregated HYRAS data? Please clarify this and if the aggregation of HYRAS applies to all related analysis then please move this detail to the methods section.

The HYRAS data have first been aggregated to the E-OBS/RCM grid of 0.22° resolution for all type of analysis in this study. We will clarify this by moving the corresponding statement more to the front of the manuscript into the method section as you have requested.

3) Further, if I understand correctly the evaluation is based on the TP1b time period (1950-2017), however the HYRAS data is only available for the period 1951-2006. Please comment on why the analysis wasn't based on the shorter HYRAS time period. I would recommend doing the analysis for 1951-2006.

In this case, you are right with the different time periods. We assume that there will be only small changes when reducing TP1b to the HYRAS period 1951-2006, but nevertheless, we will fix this for all analyses in terms of consistency.

4) The evaluation on such a highly aggregated level poses a risk of error compensation. It might be better to do the evaluation for each grid cell first (e.g. calculating the RMSE) and afterwards averaging the error metric.

We will restructure the evaluation part and will introduce additional skill measures in the evaluation. However, as the focus of this study is on intensive areal precipitation, we do not want to add too detailed grid point based analyses and take a deeper look in the spatial mean precipitation statistics. Therefore, we add additional quantities like the linear error in probability space (e.g. Potts et al., 1996), and we perform a frequency analysis on different time scales.

**5) Based on the concerns above, I don't really agree with your first point in the conclusion 'Extreme precipitation is well represented in the LAERTES-EU [P20-L352]'.**

You and also RC1 have the same concerns about this first conclusion. Thinking about this a second time, we came along that this statement might be too general. It was meant that heavy precipitation is consistent in all parts of LAERTES-EU and that our results fit in the range of previous studies (e.g. Früh et al., 2010) and also in the range of observations knowing that the used observational data sets have uncertainties as well. We will rewrite this to be more precise.

. . . Added-Value:

6) Regarding your conclusions on the IPCs showing '[. . .] a clear added value of RCM data compared to coarser global models'. From that one figure I don't really see the added value, since you haven't compared the driving GCM with RCM simulation. You have compared the IPC's to the 20CR reanalysis dataset. Because of the spatial averaging over such a large area, it might be that the trends in the GCM and RCM might be very close to each other.

**To substantiate this statement we will include the IPCs of the MPI-ESM model in Fig. 2. At least we will include the IPC of data block 1 which used the LR version of MPI-ESM, and data block 3 which used the HR version as global forcing.**

**... CC-scaling**

7) Your conclusion on the trends following the CC-scaling in your conclusion [P21-L379ff] are flawed. If you make a statement like this, please perform the temperature scaling with the LAERTES-EU temperature data and not by relating the precipitation change to a temperature approximation from another study. Please see Kröner et al (2017) and Pfahl et al (2017) for other effects than thermodynamics. Kröner et al (2017), Climate Dynamics, https://doi.org/10.1007/s00382-016-3276-3 Pfahl et al (2017), Nature Climate Change, https://doi.org/10.1038/nclimate3287

We agree with the reviewer that a relationship should be established using LAERTES-EU temperature data. We will do some brief analysis with the block 1 & 3 temperature data and put them into the context of the already cited studies on 20th century temperature changes. Then the argumentation should be more consistent and reasonable. This will be concentrated in the conclusions.

Minor comments:

P2-L25f: see also Zhang et al (2017) for a discussion on CC scaling Zhang et al (2017), Nature Geosciences, DOI: 10.1038/NGEO2911

P2-L27-34: Please add some more recent literature on this topic, e.g. Fischer and Knutti (2016), Nature, DOI: 10.1038/NCLIMATE3110 Alexander (2016), Weather and Climate Extremes, http://dx.doi.org/10.1016/j.wace.2015.10.007 Barbero et al (2017), GRL, doi:10.1002/2016GL071917

P2-L45f: Please add a view more recent studies on trends in European floods. E.g.: Blöschl et al (2017), Science, DOI: 10.1126/science.aan2506; Blöschl et al (2019), Nature, https://doi.org/10.1038/s41586-019-1495-6

P2-L49f: Connection of Heavy Precipitation over central Europe and cyclones, see also Hoffstätter et al (2017), Int. Journal of Climatology, https://doi.org/10.1002/joc.5386

P2-L55f: Also see van der Wiel et al (2019) and Martel et al (2019) for the added value of large ensembles for flood risk or return periods of heavy precipitation van der Wiel et al (2019), GRL, https://doi.org/10.1029/2019GL081967 Martel et al (2019), Journal of Climate, https://doi.org/10.1175/JCLI-D-18-0764.1

**Thank you very much for these recent references. We will go through them and decide which one and where to include them in the new version of the manuscript.**

P4-L114f: Please elaborate more on what you mean by '[. . .] more or less independent simulations'

We agree that this formulation is inept and we will remove it. What was meant is that the temporal evolution of the day-to-day weather in hindcasts is independent after a few weeks. The statement did not refer to the model system. The ensemble does not cover the full range of uncertainty, namely the model uncertainty. But, in the context of the paper we regard this as an advantage, since the data set is homogeneous over time due to the consistent model setup.

P5-L139: What do you mean by un-initialized? Please clarify this for the reader, that by initialized you mean initialized by observational(-like) salinity and other variables, whereas the un-initialized data originate from a normal CMIP5 historical simulation. I had to go to Marotzke et al (2016) to understand what was meant by this.

**We agree that this has been explained insufficiently and will state it more clearly in the revised manuscript.**

P6-L157f: Are the 99th and 99.9th percentile based on all days or wet-days only? If you want to look at heavy precipitation it might be better to look at wet days only. Like this the values would not be affected by the dry-day adjustment as much. Further, it is not clear to me if you have first spatially aggregated and then calculated the percentiles, or the other way around. Please comment on whether you think that this will make a difference to your results. This could maybe also solve your concerns on P15-L282f '[. . .] an overestimation of precipitation [...], could be a result of missing data for the applied dry-day correction.'

The percentiles were estimated using all days including dry ones. As we focus on heavy precipitation we agree with you that using wet days only would be more appropriate. Nevertheless, the uncertainty in the first half of the century will remain. The dry-day correction adjusts the number of days without precipitation (R<0.1mm) solely and not the values themselves (R>=0.1mm). This means that the dry-day correction effects the percentiles anyway in some case. But, in order to get a more thorough analysis of heavy precipitation, we will change to a wet days only calculation. Regarding your second point, we first did a spatial aggregation of precipitation to receive the areal precipitation and than calculated the percentile of this spatial mean values which are of deep interest in this study. The other way round we would get a spatial mean value of the percentile which is more relevant for a 2D analyses and related spatial variability giving local effects. We will include some sentences on that in the manuscript.

P7-L189: Could you briefly comment on why you chose the old 1961-1990 period as your reference climatology.

A couple of studies (e.g. Cahill et al., 2015 or Folland et al., 2018) showed that the climate change signal at least for global mean temperature is significantly increased since the early 1980s, which is to a lower degree applicable for Europe, too (Folland et al., 2001). Therefore, using the time period 1981-2010 would possibly include a strong changing signal to the analysis. Using 1961-1990 reduces the influence of these effects as this period shows more stable conditions to a certain degree. Doing so, there is more room for the interpretation of the future projection instead of comparing them to the directly preceding time period.

References:

Cahill et al. (2015), DOI: 10.1088/1748-9326/10/8/084002 Folland et al. (2018), DOI: 10.1126/sciadv.aao5297 Folland et al. (2001), DOI: 10.1029/2001GL012877 P9-L218f: Your conclusion to Table 2 stating that there is a higher correlation when driven by MPI-ESM-HR versus lower resolution MPI-ESM-LR is technically correct, however the differences are so marginal that I find it difficult to attribute the differences to resolution of driving data. Especially, when not only the resolution is different in the HR and LR simulations, but also the initialization. Maybe add a short sentence ['However, differences are only marginal.'].

**Thanks for that comment. Yes, the differences are marginal but the differences between the LR and HR blocks are larger than those within the LR blocks or within the HR blocks. We will include a statement such as the suggested one to the manuscript.**

Chapter 4.3: This is a nice analysis that shows the benefit of large ensembles, however since you are not looking at return values afterwards it could be nice to highlight another strength of large ensemble namely isolating the forced response from internal variability. Since you are looking at trends and variability this could be a better fit. But this is just a suggestion to improve the flow of the paper. Like a said it is a nice analysis as is.

We decided to use return values in this case as it easy to estimate and on a statistical perspective the amount of data has a significant influence on the estimates. Although you like the presented analyses, we would like to change Fig. 5 a little bit. For this particular figure the simulations were put together starting with block 1 simulation 1 and ending up with the last simulation in block 4. Doing so the shape of the curves strongly depend on the length of the single simulation runs. Therefore, we want to change the figure using the mean values of 100 random combinations of the simulation runs. The values of the signal-to-noise ration will not change that much and the given statements remain valid.

Figure 8a: Shouldn't there be also some more positive anomalies in the climTP period? Did I miss something? Because if you base the annual anomalies on this period, shouldn't you be having positive and negative anomalies within this period?

We are sorry, but unfortunately there was a wrong figure included at this point. Of course there should be and there are positive and negative anomalies in the climTP period. We replace the current plot with the correct one.

Figure 9: Nice plots! Thank you!

P22-L396f: '[. . .] can be used as input for hydrological modeling'. In general, yes and especially when looking at higher return levels of floods. However, as mentioned a few lines above this ensemble is restricted by temporal homogeneity, which can play a very important role in hydrology.

In general, we agree with that and it definitely makes sense when investigating historical events, trends, flood frequency, and so on. In a particular application and as mentioned in the following sentence, LAERTES-EU serves as stochastic weather generator which leads to quasi stochastic hydrological simulations covering the internal climate variability and also a wider range of values occurring. For such statistical applications, LAERTES-EU can be used to get robust hydrological statistics, too. For this specific case, it is necessary to do a bias correction to avoid too high discharges as a consequence of an overestimation of precipitation. This application as

well as the bias correction will be part of a consecutive study (Kautz et al., planned submission in spring 2020).

Technical Corrections:

Table 1: Projections for the period 2020-28 are missing

The projections are included in block 4 as the given year stand for the initialization years of the decadal simulations meaning an initialization in 2018 includes data for 2019-2028. We will change Table 1 accordingly so that it is clear that 'period' means the covered years.

Figure 2, 3: Please add the years of the period ([. . .] TP1b (1950-2017)). I had to go back and look for the TP1b definition. But I would anyway suggest changing the period to 1951-2006 (see major comments).

Both comments will be implemented in the revised version.

P3-L69: Typo ('Regionla', 'Regional') P5-L128: grammar (replace 'it' with 'the') P20-L342: grammar ('estimate' instead of 'estimated') Thanks! We will fix that.

Figure 4: Please clarify what the RCM spread is. I assume it to be Min-May, right?

That's true. The RCM spread means the range between the minimum and maximum occurred value of the displayed variable. We will include a short clarification into the text.

**Long-term Variances of Heavy Precipitation across Central Europe using a Large Ensemble of Regional Climate Model Simulations**

Florian Ehmele1, Lisa–Ann Kautz1, Hendrik Feldmann1, and Joaquim G. Pinto1

1Institute of Meteorology and Climate Research, Department Troposphere Research (IMK–TRO), Karlsruhe Institute of Technology (KIT), Hermann–von–Helmholtz–Platz 1, 76344 Eggenstein–Leopoldshafen, Germany. **Correspondence:** Florian Ehmele (florian.ehmele@kit.edu)

Abstract. Widespread flooding events are among the major natural hazards in Central central Europe. Such events are usually related to intensive, long-lasting precipitation over larger areas. Despite some prominent floods during the last three decades (e.-g. 1997, 1999, 2002, and 2013), extreme floods are rare and associated with estimated long return periods of more than 100 years. To assess the associated risks of such extreme events, reliable statistics of precipitation and discharge are required. Comprehensive observations, however, are mainly available for the last 50–60 years or less. This shortcoming can be reduced

using stochastic data sets. One possibility towards this aim is to consider climate model data or extended reanalyses.

This study presents and discusses a validation of different century-long data sets, a large ensemble of decadal hindcasts, and also projections predictions for the upcoming decade - Global reanalysis combined to a new large ensemble. Global reanalyses for the 20th century with a horizontal resolution of more than 100 km have been dynamically downscaled with

- 10 a regional climate model (COSMO–CLM) towards a higher resolution of 25 km. The new data sets are first filtered using a dry–day adjustment. The simulations show a good agreement with Evaluation focuses on intensive widespread precipitation events and related temporal variabilities and trends. The presented ensemble data is within the range of observations for both statistical distributions and time series. Differences mainly appear in areas with sparse observation data. The temporal evolution during the past 60 years is well captured. The results reveal some long-term variability with phases of increased
- 15 and decreased heavy precipitation precipitation rates. The overall trend varies between the investigation areas but is mostly significant. The projections predictions for the upcoming decade show ongoing tendencies with increased precipitation for upper percentiles areal precipitation. The presented RCM ensemble not only allows for more robust statistics in general, in particular it is it is also suitable for a better estimation of extreme values.

**1 Introduction**

5

20 Ongoing climate change affects not only the global scale but also impacts the regional climate. Regarding air temperature, there is a more or less clear trend in the recent past, which reveals a clear anthropogenic signal. However, various climate simulations show distinct spatial differences for precipitation trends, especially for heavy precipitation (e. g. Moberg et al., 2006; Zolina et al., 2008; Toreti et al., 2010). What is known is a theoretical increase of the water vapor eapacity according to the Clausius-Clapeyron (CC) equation of about 6–7% per degree of temperature increase

- 25 (e. g. Trenberth et al., 2003; Berg et al., 2009). For instance, Lenderink et al. (2011), Berg et al. (2013), or O'Gorman (2015) showed that this CC rate can be surpassed up to a factor 2 (Super-Clausius-Clapeyron scaling). In contrast, Stephens and Ellis (2008) found a change of precipitation below the theoretical CC rate. Nevertheless, the CC rate generally thought to be a good proxy for future precipitation projections (Westra et al., 2013).
- Easterling et al. (2000) showed that a linear trend in heavy precipitation varies for different countries and depends also on
  the considered time period. Moberg and Jones (2005) evaluated observational data from about 80 rain gauges in central and western Europe for the time period 1901–1999 revealing an increase in extreme winter precipitation. A recent (e.g. Moberg et al., 2006; Zolina et al., 2008; Toreti et al., 2010). A review of observed variability and trends in extreme climate events states that it is difficult to find significant relations between the greenhouse gas-enhanced climate change and increases or decreases in extreme precipitation events (Field et al., 2012). This is attributed to their rare occurrence, the general
- 35 high spatial variability of precipitation, and due to a lack of long-term high-quality observations. Feldmann et al. (2013) found an increase of both areal mean precipitation

Magnitude and sign of heavy precipitation trends strongly depend on various factors such as the regarded area or the considered time period (e.g. Easterling et al., 2000). Global tendencies towards more intense precipitation throughout the 20th century were revealed, for example, by Donat et al. (2016). Varying regimes between summer and winter season also account

- 40 into precipitation trends. For example, Moberg and Jones (2005) found an increase in winter precipitation across central and western Europe between 1901 and extremes in central Europe in order of 5–10% which will continue with almost same magnitude for the next decades. Moreover, the 1999, while Pal et al. (2004) found a decrease in summer precipitation for the period 1951–2000. Dittus et al. (2016) found an increasing trend between 1951 and 2005 in extreme total precipitation amounts for Europe in GCM simulations (CMIP5). Similar trends were found in global reanalyses (e.g. ERA–20C, Poli et al., 2016),
- 45 but not in observations. In contrast, Primo et al. (2019) found positive trends for two ground-based observational stations in Germany using extreme precipitation indices.

Model resolution is another crucial factor. The use of high resolution regional climate models (RCM) instead of global data sets revealed a more detailed and orographically related spatial structure of the precipitation fields and trends - Global tendencies towards more intense precipitation throughout the 20th century were also revealed by Donat et al. (2016).

- 50 In summary, these studies partly document contrasting results. Following Field et al. (2012), this can have different reasons. One major point are the underlying choice of data sets (model runs, reanalysis, and/or observations). The definition-(e.g. Feldmann et al., 2013). An increase of both areal mean precipitation and extremes in central Europe in order of 5–10% was found in RCM simulations by Feldmann et al. (2013), which will continue with almost same magnitude for the next decade. Differences in precipitation trends also stem from varying definitions of extreme events varies between such as cer-
- 55 tain thresholds, percentile-based indices, or return periods (e. g. Maraun et al., 2010). Other crucial points are that different time periods and areas were investigated as well as different model resolutions(e.g. Maraun et al., 2010). While most of these studies show trends in daily precipitation, just a few deal with sub-daily trends. Barbero et al. (2017), for instance, compared trends in sub-daily and daily extremes. Although significant increasing trends were found for both time ranges, trends in daily extremes are better detected than in sub-daily extremes.

- 60 Spatially extended intensive rainfall events are frequently related to widespread flooding along the main river networks of central Europe causing major damage in the order of several billion euro (EUR) per event (e.g. Uhlemann et al., 2010; Kienzler et al., 2015; Schröter et al., 2015; MunichRe, 2017). Mudelsee et al. (2003) investigated the trends in the occurrence of extreme floods related to heavy precipitation events along the Oder and Elbe rivers. They found a decrease for winter floods in both river catchments, while there seems to be no significant trend for summer floods. In contrast,
- 65 Dittus et al. (2016) found an increasing trend between 1951 (e.g. Uhlemann et al., 2010; Kienzler et al., 2015; Schröter et al., 2015; MunichRe, 2017). A prominent example of such an extreme and 2005 in extreme total precipitation amounts for e.g. Europe in global climate model simulations (CMIP5). Similar trends were found in reanalyses (e.g. ERA–20C, Poli et al., 2016), but not in observations. Moreover, Mudelsee et al. (2004) and Nissen et al. (2013) highlighted a strong dependency of central European flood events on the specific weather pattern of cyclone
- 70 pathway "Vb" like the severe flood event of 2002 devastating event is the flood in 2012 along the rivers Elbe and Danube (Ulbrich et al., 2003a, b). Such outstanding events are by definition extremely rare, which makes the risk estimation difficult or althe limited most impossible due to time period with available area-wide observations (e.g. Pauling and Paeth, 2007; Hirabayashi et al., 2013). Nevertheless, the estimation of flood risk and related trends for (e.g. Pauling and Paeth, 2007; Hirabayashi et al., 2013). However, trend analyses of such extreme events and the related risks
- 75 during the past and for the future are of great importance for insurance purposes or flood protection (e.g. Merz et al., 2014; Schröter et al., 2015; Ehmele and Kunz, 2019).
  (e.g. Merz et al., 2014; Schröter et al., 2015; Ehmele and Kunz, 2019).

A possible way of dealing with the unsatisfactory data availability are century-long simulations using climate models (e. g. Stucki et al., 2016) (e. g. Stucki et al., 2016) or stochastic approaches 80 (e. g. Peleg et al., 2017; Singer et al., 2018; Ehmele and Kunz, 2019).

Several previous studies have investigated long-term trends and variability of extreme precipitation using century-long reanalysis data sets. For instance, (
[revised manuscript text omitted]
 consists of a combined large downscaling ensemble of simulations with one RCM. There are two different types: long-lasting simulations of 45–110 years and simulations over one decade. In the latter, only a period of 10 years (e.g. 1961–1970) was simulated with a specific number of ensemble members. Then, the initialization point was shifted by one

- 165 year (e.g. 1962–1971) and so on until the end of the covered time period. In total, LAERTES-EU consists of 1183 more or less independent simulations (sample size) with approximately 12.500 simulated years. The number of ensemble members at a specific time varies from 6 at the beginning of the century to a maximum of 188 members between 1970 and 2000 (see Fig. S1 in Supplementary).
- LAERTES-EU is divided into four different data blocks (Table 1). All regional simulations used combines a large number of
   regional dynamical downscaling simulations for Europe performed with a single RCM. The used RCM is the non-hydrostatic model of the Consortium for Small-scale Modeling Modelling (COSMO) in climate mode model version 5 (CCLM5; Rockel et al., 2008)and have, which has a spatial resolution of 0.22° (≈ 25 km). The model covers the EURO-CORDEXEURO-CORDEX1 domain (Jacob et al., 2014). All the simulations were Overall, the simulations use the same domain, model version and set-up, which was adapted from EURO-CORDEX (Kotlarski et al., 2014). According to Feldmann et al. (2008), a dry–day correction
- 175 is important as climate models tend to overestimate the number of wet days with low intensities below 0.1 mm, known as the drizzle effect (Berg et al., 2012). In order to reduce this typical bias, a dry-day adjustment was first applied to LAERTES-EU. The E-OBS data were used for this correction, as they have the same spatial extension and resolution as the CCLM simulations. All simulations are performed within the BMBF (Federal Ministry of Education and Research of Germany) project MiKlip II2 (Marotzke et al., 2016) For all simulations the same domain, model version and set-up, adapted from EURO-CORDEX, were
   180 used.

The boundary forcing was to create and test a decadal prediction system including a regional downscaling component for Europe.

For all downscaling simulations the boundary conditions were derived from the Max–Planck Institute of Meteorology coupled Earth System Model (MPI–ESM). This global model consists of the atmospheric component ECHAM6 (Stevens et al., 2013), the ocean component MPI–OM (Jungclaus et al., 2013), and the land-surface model JSBACH (Hagemann et al., 2013).

LAERTES-EU is divided into four different data blocks (Table 1) depending on the setup of the forcing MPI–ESM ensemble simulations. The differences between the four different data blocks stems from the setup, external forcing and initialization of the MPI–ESM simulations. The data blocks 1 and 2 of the RCM ensemble (cf. Table 1) obtained it the boundary values from the MPI–ESM–LR simulations using a T63 resolution and 47 vertical layers. Data block 3 and 4 used the MPI–ESM–HR

<sup>1http://www.euro-cordex.net

<sup>2https://www.fona-miklip.de/

**Table 1.** Overview of the RCM ensemble LAERTES-EU with the name of the simulation within the MiKlip project, the classification into data blocks, the underlaying set-up (experiment), the covered time period, and the number of simulation years. For data blocks 2 and 4, period means the range of the initialization years; XX stand for the ensemble number and YYYY for the initialization year.

| name                 | block | experiment             | period                                | years | comment                                |
|----------------------|-------|------------------------|---------------------------------------|-------|----------------------------------------|
| as20ncepXX           | 1     | 20CR via MPI-ESM-LR    | 1900–2009                             | 330   | 3 members of 110 years each            |
| decXXoYYYY           | 2     | MPI-ESM-LR DROUGHTCLIP | <del>1910–2009</del> 1911–2019 | 3000  | 3 members with 100 decades each        |
| historical_rXi1p1-HR | 3     | MPI-ESM-HR HISTORICAL  | 1900–2005                             | 410   | run 1–3 each with 106 years,           |
|                      |       |                        |                                       |       | run 4–5 each with 46 years (1960–2005) |
| preop                | 4     | MPI-ESM-HR CMIP5       | <del>1960-2016-1961-2026</del> | 2850  | 5 members with 57 decades each         |
| dcppA-hindcast       | 4     | MPI-ESM-HR CMIP6       | <del>1960-2018-</del> 1961-2028       | 5900  | 10 members with 59 decades each        |

- 190 version (Müller et al., 2018) as their driving model. In this version, the horizontal resolution is T127 and 95 vertical layers are applied. Three types of forcing ensembles can be distinguished:
  - (I) MPI–ESM assimilates reanalysis data for long-term simulations (data block 1).
  - (II) Long-term historical-type simulations, according to the CMIP5 specifications (data block 3; Taylor et al., 2012).
  - (III) Initialized decadal (10-year) hind- and forecast simulations (data blocks 2 and 4).
- 195 The MPI-ESM forcing data used for the three long-term simulations in data In data block lassimilated the, the first type (I) is applied. Here the 20th Century Reanalysis (20CR; Compo et al., 2011; Müller et al., 2014) over the period 1900–2009data (20CR; Compo et al., 2011) are assimilated into the MPI-ESM-LR (Müller et al., 2014). 20CR has a spatial resolution of approximately 2° (T62) and was generated using the Global Forecast System (GFS; Kanamitsu et al., 1991; Moorthi et al., 2001) of the National Centers for Environmental Prediction (NCEP)3. It used a 56 member Ensemble Kalman Filter approach
- to assimilate surface pressure, monthly sea surface temperature and sea-ice observations. From these simulations the starting conditions for a decadal hindcast ensemble (data block 2) has been derived (Mieruch et al., 2014; Müller et al., 2014; Reyers et al., 2019; Feldmann et al., 2019). Each year three initialized decadal simulations were started, to study the long-term predictive skill on decadal time scales. Three of the 20CR members are assimilated into MPI-ESM to provide long-term (110 years each) climate reconstruction simulations over the period 1900–2009 (Müller et al., 2014).
  Afterwards, a downscaling with CCLM uses these global simulations as boundary conditions (e.g. Primo et al., 2019).
- Afterwards, a downscaling with CCLM uses these global simulations as boundary conditions (e.g. Primo et al., 2019). Data block 3 contains the downscaling of five un-initialized (historical)consists of the second type (II), were five so called historical simulations of MPI–ESM–HR with CMIP5 observed natural and anthropogenic external forcing (Taylor et al., 2012).

<sup>3http://www.ncep.noaa.gov/

Data block climate forcing (Taylor et al., 2012) are used as boundary conditions for CCLM. The ensemble was generated by starting the MPI–ESM from arbitrary dates in a pre-industrial control simulation (Müller et al., 2014). Three of the five CCLM

210 members cover the period 1900–2005 (106 years each). The two additional simulations cover the period 1960–2005 (46 years each).

Data block 2 and 4 encompasses two sets of decadal hindcasts over the period since 1960 (Müller et al., 2012; Marotzke et al., 2016). The preop-ensemble consist of initialized decadal simulations (type III). The starting conditions are derived from an observed state (Müller et al., 2012; Marotzke et al., 2016). For each starting year, an ensemble of decadal simulations is generated and

- 215 then, the initialization point is shifted by one year (e.g. 1961–1970, 1962–1971, and so on). Due to the overlap, a specific calendar year may be covered by several decadal hindcasts with different starting years. These decadal hind- and forecasts thus represent the current state of the major modes of climate variability compared to the so-called un-initialized historical simulations (data block 3). The downscaling procedure, the skill, and the added value are described in Mieruch et al. (2014), Feldmann et al. (2019), and Reyers et al. (2019).
- 220 In data block 2, the starting conditions of the three decadal hindcast members with MPI–ESM–LR are derived from the assimilation experiments in data block 1. The starting years of the CCLM downscaling range from 1910 to 2009. This means the last simulated year is 2019.

Data block 4 consists of two parts. Both of them use the MPI–ESM–HR version. The so-called preop-ensemble has five memberseach year. The climate forcing for these simulations stems also. The external climate forcing is derived from CMIP5;

- 225 whereas for the 10 member per year deppA-ensemble the CMIP6 external forcing was applied (Eyring et al., 2016; Boer et al., 2016). . The starting years range from 1960 to 2016 (last simulated year 2026). The so-called dcppA-hindcast ensemble has ten members and uses the external forcing for CMIP6 (Eyring et al., 2016). The global simulations are a contribution to the Decadal Climate Prediction Project of CMIP6 (DCPP; Boer et al., 2016). The starting years are 1960 to 2018 (last simulated year 2028).
- The In total, LAERTES-EU consists of 1183 simulation runs (sample size) with approximately 12.500 simulated years. The number of ensemble members for a specific year varies from six at the beginning of the century to a maximum of 188 members between 1970 and 2000 (see Fig. S1 in the supplemental material). The simulation in all four data blocks are affected by the observed external climate forcing, but they differ with respect to the representation of the observed climate variability, whereas data block 1 uses assimilated 20CR reanalysis data, data block 2 and 4 contain initialized hindcastsand, which to some
- 235 degree follow the observed low frequency variability, and data block 3 only uses the external forcing information. Nonetheless, the four groups of downscaling simulations can be grouped into a large ensemble, since the regional simulations were all performed with the same setup of the RCM. Despite the same initial conditions and model setup, the temporal evolution of the day-to-day weather is (statistically) independent between the members after a few weeks. This is an advantage, since the data set is homogeneous over time but also covers uncertainties in the observations including unknown and not yet observed events. The validity of this combination approach is tested within Sect. 4.
  - In order to reduce well-known limitations of climate model simulation, the ensemble data first were filtered using a dry-day adjustment. According to Feldmann et al. (2008), a dry-day correction is essential as climate models tend to overestimate the

number of wet days with low intensities below 0.1 mm (Berg et al., 2012), known as the drizzle effect. The dry-day correction was performed using the E-OBS data, as it has the same spatial extension and resolution.

[revised manuscript text omitted]

**Figure 1.** Topographic map of Europe at model resolution 0.22° (in meters above mean sea level; m a.m.s.l.) with the PRUDENCE regions Mid-Europe (ME; dark red box) and the Alps (AL; gray box), state borders (black contours), and the HYRAS area (light red contour). Ocean grid cells are set to a missing value.

significantly increased since the early 1980s. Therefore, using the time period 1981–2010 as reference would possibly include
a strong changing signal to the analysis. Using 1961–1990 reduces the influence of these effects, as this period shows more
stable conditions to a certain degree. This also permits more room for the interpretation of the future predictions.

310

**4 Validation of the RCM ensemble**

In the following, the above described methods are applied in order to validate LAERTES-EU concerning its representativeness with observations. With this aim, data for the investigation period TP1b is used - and the boxes ME and AL (cf. Fig. 1) are limited to the HYRAS area (ME\* and AL\*).

**315 4.1 StatisticsStatistical distributions and frequencies**

The IPCs give the range of simulated (observed) precipitation intensities at any grid point in-within the investigation area and its corresponding probability (Fig. 2). For both investigation areas, the IPCs reveal a distinct added value of the RCM compared to the global model. Due to the coarse resolution, the GCM is incapable of simulating-intensities greater than approximately  $60100 \text{ mm d}^{-1}$  and underestimates are not found in the GCMs, which underestimate by a large degree the probability of a wide

320 range of intensities. the high intensities. The same applies for the global reanalysis 20CR. On the other hand, the RCM tend to